# Towards LLM4QPE: Unsupervised Pretraining of Quantum Property Estimation and A Benchmark

**Yehui Tang[1], Hao Xiong[1], Nianzu Yang[1], Tailong Xiao[23], Junchi Yan[1]***

[1]Department of Computer Science and Engineering, Shanghai Jiao Tong University
[2]Institute of Quantum Sensing and Information Processing, Shanghai Jiao Tong University
[3]Hefei National Laboratory, Hefei, China
{yehuitang,taxuexh,yangnianzu,tailong_shaw,yanjunchi}@sjtu.edu.cn

## Abstract

Estimating the properties of quantum systems such as quantum phase has been critical in addressing the essential quantum many-body problems in physics and chemistry. Deep learning models have been recently introduced to property estimation, surpassing conventional statistical approaches. However, these methods are tailored to the specific task and quantum data at hand. It remains an open and attractive question for devising a more universal task-agnostic pretraining model for quantum property estimation. In this paper, we propose LLM4QPE, a large language model style quantum task-agnostic pretraining and finetuning paradigm that 1) performs unsupervised pretraining on diverse quantum systems with different physical conditions; 2) uses the pretrained model for supervised finetuning and delivers high performance with limited training data, on downstream tasks. It mitigates the cost for quantum data collection and speeds up convergence. Extensive experiments show the promising efficacy of LLM4QPE in various tasks including classifying quantum phases of matter on Rydberg atom model and predicting two-body correlation function on anisotropic Heisenberg model.

## 1 Introduction

Estimating quantum system properties such as quantum phase is essential for verifying and evaluating quantum technologies (Huang et al., 2020; Gočanin et al., 2022), which is often in the form of many-body problems. Precise estimation of generic quantum systems is challenged due to the exponential complexity inherent in describing quantum many-body systems (Gebhart et al., 2023). Fortunately, physical systems of interest such as those generated by the dynamics of local Hamiltonians are not generic, since their particular structure guarantees that the full complexity of Hilbert space is in principle not required for their accurate description (Carrasquilla et al., 2019). Accordingly, statistical (including learning-based) approaches have emerged to characterize quantum systems from traditional Density Functional Theory (DFT) (Hohenberg & Kohn, 1964), Quantum Monte Carlo (QMC) (Ceperley & Alder, 1986), to advanced variational methods e.g. Tensor Networks (TNs) (Orús, 2019) and Neural Network Quantum States (NNQS) (Zhang & Di Ventra, 2023).

There are basically two categories of variational methods for quantum property estimation (QPE). The first category refers to the TNs and NNQS which formulate QPE as an optimization problem where the quantum state is approximately represented by a parameterized wave function. The parameterized wave function is updated by minimizing the expectation values of relevant observable estimators, based on either density matrix renormalization group (DMRG) algorithm (White, 1992) or variational Monte Carlo (VMC) (McMillan, 1965). Afterwards the interested properties can be analyzed by preforming algebra operations on the wave function. Another line of research resorts to neural networks to serve as universal functions for directly approximating quantum system properties (Gilmer et al., 2017; Kawai & Nakagawa, 2020; Xiao et al., 2022), which we call NNQPE. The input to the neural networks is the measurement results of the quantum state, and the output is the

---

*Correspondence author. Work was partly supported by NSFC (62222607), Shanghai Municipal Science and Technology Major Project (2021SHZDZX0102), SJTU Trans-med Awards Research (STAR) 20210106.

property of interest. The parameters are optimized using gradient descent. The goal of NNQPE is to accurately characterize the properties of the quantum state using as few identical copies and measurements as possible. Compared with the TNs, this class of methods could more easily display non-local correlations, allowing in principle to capture quantum states with higher entanglement (Huang et al., 2022). Moreover, rather than TNs and the NNQS where additional computational overheads is required to extract the properties given the optimized parameterized wave function, NNQPE can directly predict the properties for unknown quantum states.

However, NNQPE suffers generalization ability issue, especially given limited measurement data for training (Gebhart et al., 2023). Although the generalizability could be improved by training the models based on extensive measurement data and corresponding labels, the labeling process, i.e., accurately estimating properties of quantum systems requires computational and memory resources that increase exponentially with the system size (Carleo et al., 2019). In particular, the labeling efforts for quantum systems are intensive. For example, DFT suffers from self-interaction error and delocalization error, making it difficult to represent quantum states with strong correlations (Verma & Truhlar, 2020). The sign problem (Loh Jr et al., 1990) implies that it is intractable for QMC to evaluate properties for large systems or systems with low temperatures (Troyer & Wiese, 2005; Huang et al., 2022). The maximum bond dimensions of TNs for precisely preserving the properties of quantum states such as the entanglement entropy scales exponentially w.r.t. the evolution time (Brandao & Horodecki, 2015). In conclusion, the labeling process is hard to complete classically due to the inherent separation between quantum and classical computing.

Furthermore, despite the significant promise of NNQPE, their application in harnessing advanced machine learning techniques for quantum physics remains in its early stages. Current models of NNQPE are tailored and trained for particular quantum systems and specific tasks. This approach contrasts sharply with the era of Large Language Models (LLMs) (Radford et al., 2018; Brown et al., 2020), which have achieved general-purpose language generation and understanding capabilities. In the realm of LLMs, pretraining serves as the primary method for capturing general language understanding and afterwards finetuning is adopted to adapt the model to accomplish specialized tasks. This distinction highlights the nascent yet evolving nature of applying sophisticated machine learning strategies within the quantum physics domain.

In fact, with the increasing scale of the quantum devices, a vast amount of quantum data are produced by quantum measurement (Brydges et al., 2019). Such data holds intricate details about the system. An open question is designing a versatile model, which undergoes extensive pretraining to master these quantum intricacies. The success of deep learning in handling high-dimensional data sheds lights on answering this question. First, the sheer volume of quantum data from measurements allows for the extraction of meaningful patterns and representations (Anshu & Arunachalam, 2024). Second, the universal approximation capabilities of neural networks suggest that given sufficient data and computational resources, it's possible to model the complex, nonlinear relationships inherent in quantum systems (Carleo et al., 2019; Gebhart et al., 2023). Lastly, the task-agnostic nature of pretraining (Liu et al., 2023) aligns with the quantum realm's diversity, where a single model can learn hidden features across various systems and physical conditions. This feasibility is further supported by the principle of transfer learning (Weiss et al., 2016), where knowledge gained in one context can significantly benefit task-specific applications.

In this paper, we introduce an **LLM**-style task-agnostic pretraining model **for Q**uantum **P**roperty **E**stimation named **LLM4QPE**. This model is pretrained by leveraging vast (unlabeled) quantum data, across diverse quantum systems of the same family govern by different physical conditions. For the downstream tasks, we finetune LLM4QPE on two typical QPE tasks including classifying quantum phases of matter and predicting two-body correlation function. We also consider two families of quantum model including the Rydberg atom model and the anisotropic Heisenberg model. The results show its promising power for tackling QPE problems especially in scenarios with limited data availability. The contributions are:

1) Departure from most existing supervised learning QPE models reliant on restricted, task-specific labeled quantum data, we propose LLM4QPE, to our best knowledge, the first LLM-style model for quantum property estimation. Its unsupervised pretraining is fulfilled by maximizing the expected log likelihood of measurement bit strings, which is entirely unsupervised and task-agnostic.

2) We develop the novel architecture of our LLM4QPE model. Specifically, to embed the batch-style discrete measurement records to a continuous space, a trainable LSTM embedding layer is attached to the transformer decoder. The LSTM-Transformer architecture provides an innate framework for handling diverse quantum data stemming from experiments under varying physical conditions, enabling prediction of the property of quantum systems of the same family.

3) We collect a set of quantum data from simulations for unsupervised pretraining and supervised finetuning. For pretraining, the dataset consists of quantum state measurement records, the size of which scales linearly w.r.t. the system size and the number of measurements, along with the values of physical condition variables determining the evolution of quantum systems. Downstream tasks utilize a set of data generated from quantum systems of the same family, with additional system properties serving as labels for tasks like phase classification and correlation prediction.

4) We verify the superiority of our approach by empirical studies on two QPE tasks: classifying quantum phases of matter on Rydberg atom model and predicting two-body correlation function on anisotropic Heisenberg model, given limited measurements on a resource-limited device.

## 2 PRELIMINARIES OF QUANTUM STATE AND QUANTUM MEASUREMENT

We introduce basic concepts of quantum computing. Please refer to (Nielsen & Chuang, 2010) for more details. We put the details on related work to Appendix A.

**Quantum State and Density Operator.** The quantum bit named as *qubit* is the basic unit of the quantum system. We call the ensemble of all qubits in a (sub)system the *quantum state*. The qubit is in superposition and becomes deterministic once the measurement is performed on it. How a quantum state is described mathematically depends on the chosen basis state. For example, by using two orthogonal *computational basis states*[1] $|0\rangle = \left[\begin{smallmatrix} 1 \\ 0 \end{smallmatrix}\right]$ and $|1\rangle = \left[\begin{smallmatrix} 0 \\ 1 \end{smallmatrix}\right]$, one qubit can be described mathematically as a linear combination $|\phi\rangle = \alpha|0\rangle + \beta|1\rangle = \left[\begin{smallmatrix} \alpha \\ \beta \end{smallmatrix}\right]$ in the space $\mathbb{C}^2$, where $\alpha, \beta \in \mathbb{C}$ are the *amplitudes* satisfying $|\alpha|^2 + |\beta|^2 = 1$. An alternate formulation for describing the quantum state is possible using a tool known as the *density operator* or *density matrix*. For example, the density matrix of $|0\rangle$ is $\rho_0 = |0\rangle\langle 0| = \left(\begin{smallmatrix} 1 & 0 \\ 0 & 0 \end{smallmatrix}\right)$ where $\langle 0|$ denotes the conjugate transpose of $|0\rangle$. For a generic $L$-qubit quantum state, it can be described by the so called *wave function*:

$$|\psi\rangle = \sum_{\sigma_1=1}^{M} \cdots \sum_{\sigma_L=1}^{M} \mathbf{\Psi}(\sigma_1, \ldots, \sigma_L)|\sigma_1, \ldots, \sigma_L\rangle, \tag{1}$$

where $\mathbf{\Psi} : \mathbb{Z}^L \to \mathbb{C}$ maps a fixed configuration $\boldsymbol{\sigma} = (\sigma_1, \ldots, \sigma_L)$ of $L$ qubits to a complex number satisfying $\sum_{\sigma_1=1}^{M} \cdots \sum_{\sigma_L=1}^{M} |\mathbf{\Psi}(\sigma_1, \ldots, \sigma_L)|^2 = 1$, and $\sigma_i \in \{1, \ldots, M\}$ is one of the $M$ possible outcomes by performing quantum measurement on the $i$-th qubit. The wave function is formulated in a complex Hilbert space where the vector representation of the quantum state $|\psi\rangle \in \mathbb{C}^{M^L}$ and its density matrix $|\psi\rangle\langle\psi| \in \mathbb{C}^{M^L \times M^L}$, which becomes astronomical for large $L$.

**Quantum Measurement.** It converts some of the quantum information into classical form (for further processing), as described by a set of *measurement operators* $\{\mathbf{O}_m\}_{m=1}^{M}$ satisfying $\sum_m \mathbf{O}_m = \mathbf{I}$, where $M$ is the total number of operators. Measuring a qubit leads to collapse of the wave function and produces potentially yield different outcomes. The possible outcomes correspond to the indices $m$ of measurement operators. Concretely, upon measuring the qubit $\rho$, the probability of getting the result $m$ is given by $p(m) = \text{tr}(\rho\mathbf{O}_m)$. For a quantum state with $L$ qubits, the common strategy is to measure each of the qubits in *parallel* (Leibfried et al., 1996; Jullien et al., 2014). According to the born rule of quantum mechanics, such a measurement procedure outputs a measurement string $\boldsymbol{\sigma} = (\sigma_1, \ldots, \sigma_L)$ where $\sigma_i \in \{1, \ldots, M\}$ with probability $|\mathbf{\Psi}(\sigma_1, \ldots, \sigma_L)|^2$ as given in Eq. 1.

## 3 LLM4QPE

### 3.1 OVERVIEW

As shown in Fig. 1, our model involves two steps: pretraining and finetuning. For pretraining, the model is fed with unlabeled $\mathcal{D}_p$, and undergoes fully unsupervised training. Subsequently, the pretrained parameters are transferred to the supervised finetuning phase, where all the parameters

---

[1]Computational basis states are also referred to as the $Z$-basis states in some literature.

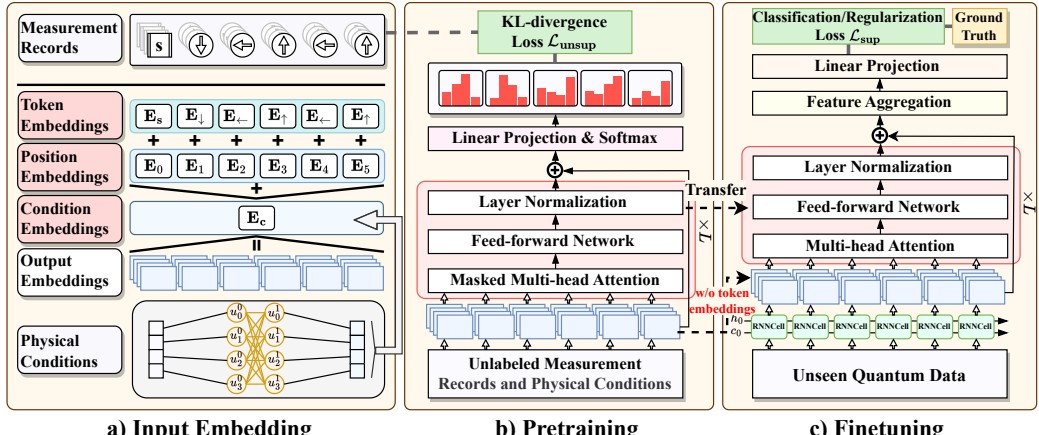

Figure 1: **Pretraining and finetuning of LLM4QPE. a)** The output embeddings are the summation of token embeddings, condition embeddings and position embeddings. Three embeddings correspond to encode discrete measurement records, continuous physical variables and qubit positions, respectively. The token embeddings are replaced with the LSTM embeddings while finetuning. **b)** The main part of the model is a multi-layer transformer decoder. Pretraining is entirely unsupervised. The output target is to approximate the *classical* distribution of the wave function. **c)** The model for finetuning and pretraining share the same structure. The pretrained parameters are transferred to the finetuning stage and updated towards a task specific supervised loss.

are updated using labeled data $\mathcal{D}_t$ for various downstream tasks with their task-specific supervised losses. Finally, we evaluate our LLM4QPE using dataset $\mathcal{D}_e$. Each downstream finetuning model possesses separate parameters, even though they initially share the same pretrained parameters. One of the most notable aspects of our model is the consistent structural similarity between pretraining and finetuning, with only a few small modifications when handling different downstream tasks.

The description of the quantum data is discussed in Sec. 3.2. We make an analogy between quantum data and text that, each measurement outcome $\sigma_i$ of a qubit is analogue to the token, and the number of the possible outcomes $M$ is likely to the vocabulary size $|\mathcal{V}|$. A measurement string $\boldsymbol{\sigma}$, which resembles the sentence in texts, is a projection of the entire quantum system with correlative effects among them. The collection of measurement records $\mathbf{R}$ comprised of many measurement strings from various physical conditions are akin to the corpus gathered from various sources and genres. In fact, these have also been mentioned implicitly in (Sharir et al., 2020; Hibat-Allah et al., 2020; Cha et al., 2021; Zhang & Di Ventra, 2023). Yet existing works are still confined to the single task for training and testing, involving no pretraining. Our model, in contrast, draws inspiration from LLMs to handle quantum data. Specifically, the data type and data collection strategies are described in Sec. 3.2 and more details can be found in Appendix B. Given the generated datasets, we first discuss how to unsupervisely pretrain LLM4QPE in Sec. 3.3. Afterwards the pretrained parameters are updated towards a supervised loss for different tasks, as presented in Sec. 3.4.

## 3.2 DESCRIPTION OF THE QUANTUM DATASET GENERATED FROM SIMULATION

We first provide the definition of the quantum dataset in Def. 1 in which the procedures of quantum dataset generation are provided. An easy-to-understand flowchart is also provided in Fig. 2.

**Definition 1 (Quantum Dataset).** *The quantum dataset is described as $\mathcal{D} = \{\mathbf{s}_i\}$. Each sample $\mathbf{s}_i = (\mathbf{R}_i, \mathbf{c}_i, \mathbf{p}_i)$ contains the measurement records $\mathbf{R}_i$, the physical condition variables $\mathbf{c}_i$ and the (optional) system property variables $\mathbf{p}_i$. Let $L$ denote the number of qubits, $K$ represent the number of copies of each quantum state and $M$ denote the number of possible outcomes by performing measurement on a single qubit. We explain their meaning in detail below.*

1) *$\mathbf{c}_i \in \mathbb{R}^C$ represents the physical condition variables controlling the evolution of the quantum system. These variables can be directly obtained when initializing quantum experiments. The types of the variables could be system size, coupling strength of Hamiltonians, etc.*

2) *The measurement records, denoted as $\mathbf{R}_i \in \mathbb{Z}^{K \times L}$, are outcomes generated by quantum measurement. A quantum state is generated by evolving the system under a fixed physical condition*

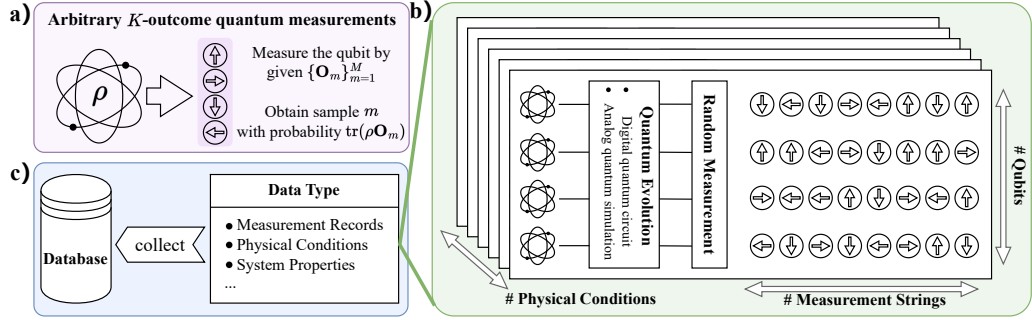

Figure 2: **Process of generating the quantum dataset.** **a)** For each qubit of the quantum system, we perform quantum measurement using operators $\{\mathbf{O}_m\}_{m=1}^M$ and obtain an integer outcome $m$ with probability $p(m)$. **b)** Consider the quantum system govern by different physical conditions. Quantum measurements are performed on an ensemble of identical quantum states evolved under each of fixed physical conditions. Measurement can be done parallel for all the qubits of single copy of the quantum state and outputs a measurement string. This process is applicable and feasible to existing digital and analog quantum computers. **c)** The collected data are structured and packed into a series of tensors, which can be efficiently stored into classical devices and easy to process.

*specialized by $\mathbf{c}_i$. Afterwards quantum measurement is performed independently on each qubit in parallel using a set of measurement operators $\{\mathbf{O}_m\}_{m=1}^M$. Performing measurement on $L$ qubits results in a measurement string, represented as $\boldsymbol{\sigma} = (\sigma_1, \ldots, \sigma_L)$ where each $\sigma_l \in \{1, \ldots, M\}$. The measurement procedures above are repeated $K$ times for each copy of the quantum state. Finally, we collect $K \times L$ measurement outcomes and store them within $\mathbf{R}_i$.*

3) *(Optional) Certain system property $\mathbf{p}_i \in \mathbb{R}^P$ represents the statistics of the quantum system conditioned on $\mathbf{c}_i$, such as the quantum phase, correlation function, entanglement entropy, purity, etc. The exact values of $\mathbf{p}_i$ can be calculated by classical post-processing by analyzing the either the wave functions or measurement statistics. We treat these properties as supervised labels which used for finetuning the model.*

It should be mentioned that the process of quantum dataset generation above is closed to Wang et al. (2022). The difference is that LLM4QPE requires additional ground-truth labels of system properties for finetuning, rather than the suggestions of Wang et al. (2022) in which the authors propose to reconstruct the quantum state by unsupervised learning on measurement records, afterwards classical shadow (Huang et al., 2020) is required to predict specific quantum properties. The two step strategy often introduces additional overheads. Furthermore, our experiments indicate that parameters in LLM4QPE are specifically optimized for corresponding objectives such as quantum phase of matters and correlation function, which often leads to superior performance in our numerical results.

### 3.3 UNSUPERVISED PRETRAINING

Unlike the previous studies (Czischek et al., 2022; Zhang & Di Ventra, 2023) which consider the pretraining as a warmup process to find suitable initialization for model's parameters and then finetune the model on the specific system with the same learning objective as pretraining. Instead, LLM4QPE regards the pretraining as the avenue to master the quantum intricacies across different systems of the same family. The pretrained parameters can be transferred towards various downstream tasks. LLM4QPE is pretrained in a fully unsupervised manner, as illustrated in Fig. 1b.

**Quantum Data for Pretraining.** The quantum dataset $\mathcal{D}_p = \{\mathbf{R}_i, \mathbf{c}_i\}_{i=1}^{N_p}$ used for pretraining is constructed using the strategy discussed in Sec. 3.2. Here we discuss how to reorganize the data to adapt to LLM4QPE's unsupervised pretraining. Let $K_p$ be the number of measurement strings used for pretraining. We stack all the input measurement records $\{\mathbf{R}_i\}_{i=1}^{N_p}$ along the first dimension and output $\mathbf{E}_{\text{in}} \in \mathbb{Z}^{N_p K_p \times L}$, where each row is a measurement string $\boldsymbol{\sigma}_b \in \mathbb{Z}^L$. We also construct the matrix $\mathbf{C}_{\text{in}} \in \mathbb{R}^{N_p K_p \times C}$ where each row is the values of physical condition variables $\mathbf{c}_b \in \mathbb{R}^C$. For both the Rydberg atom model and the anisotropic Heisenberg model, we fix $N_p = 100$ and $K_p = 1024$. For each training iteration, we randomly sample $B_p$ rows of $\mathbf{E}_{\text{in}}$ and $\mathbf{C}_{\text{in}}$. Such that the input of the model is $\{(\boldsymbol{\sigma}_b, \mathbf{c}_b) | \boldsymbol{\sigma}_b \in \mathbf{E}_{\text{in}}, \mathbf{c}_b \in \mathbf{C}_{\text{in}}\}_{b=1}^{B_p}$ with batch size $B_p$.

**Input Embeddings.** As shown in Fig. 1a, we consider three types of embeddings as input to capture the hidden patterns of the quantum system: token embeddings, condition embeddings and position embeddings. Since each element of the measurement string $\boldsymbol{\sigma}_b$ is a discrete integer $\sigma \in \{1, \dots, M\}$ which resembles to the token in NLP, we use learned embeddings to convert the measurement string $\boldsymbol{\sigma}_b$ with additional start token s and output the token embeddings $\mathbf{E}_t \in \mathbb{R}^{B_p \times (L+1) \times d}$ where $d$ is the feature dimension. We empirically find that encoding the physical condition into the model can further improve the performance. A Feed-Forward Network (FFN) with one hidden layer is used to embed the physical condition $\mathbf{c}_b$ into the feature vector $\mathbf{E}_c \in \mathbb{R}^{B_p \times d}$. It is treated as a sentence-level embedding which will be added to all of the $L$ measurement tokens, and we call it the global embedding. Subsequently, the input embeddings are the (broadcasting) summation $\mathbf{E}_{\text{out}} = \mathbf{E}_t + \mathbf{E}_c + \mathbf{E}_p$ where $\mathbf{E}_p$ is the positional embeddings as the same as (Vaswani et al., 2017). $\mathbf{E}_{\text{out}}$ is then processed by deeper layers in the discussion below.

**Model Architecture.** As depicted in Fig. 1b, the main part of LLM4QPE is a multi-layer transformer decoder which originates from (Vaswani et al., 2017). The input is the embedding $\mathbf{E}_{\text{out}}$ and the output is $\mathbf{H} \in \mathbb{R}^{B_p \times (L+1) \times d}$, which are high-order representations of all the measurement strings and the conditional variables in a batch. Please refer to (Vaswani et al., 2017) for more details on transformer. For pretraining, given a fixed qubit configuration $\boldsymbol{\sigma} = (\sigma_1, \dots, \sigma_L)$, LLM4QPE attempts to approximate the *classical* distribution $p(\sigma_1, \dots, \sigma_L) = |\boldsymbol{\Psi}(\sigma_1, \dots, \sigma_L)|^2$ in Eq. 1. Such joint distribution is approximated by factorizing it into a product of conditional probabilities:

$$p(\sigma_1, \dots, \sigma_L | \mathbf{c}) = \prod_{l=1}^{L} p(\sigma_l | \sigma_{l-1}, \dots, \sigma_1, \mathbf{c}). \tag{2}$$

The parameters are optimized by minimizing the average negative log-likelihood loss:

$$\mathcal{L}_{\text{unsup}} = \frac{1}{B_p} \sum_{(\boldsymbol{\sigma}, \mathbf{c}) \in \mathcal{D}_p} -\log p(\sigma_1, \dots, \sigma_L | \mathbf{c}), \tag{3}$$

which corresponds to the maximization of (conditional) likelihoods concerning the observed measurement outcomes. Pretraining is entirely unsupervised, enabling the model to be trained on extensive quantum data that encompass a wide range of physical conditions. To maintain the physical validity that restricts the output distribution to be normalized, a general strategy is employed to fix the last layer as the linear projection with *softmax* activation function, such that the output distribution satisfies $\sum_{\sigma_1=1}^{M} \cdots \sum_{\sigma_L=1}^{M} p(\sigma_1, \dots, \sigma_L) = 1$ (see Appendix C for proof).

## 3.4 SUPERVISED FINETUNING

The self-attention mechanism in the transformer allows LLM4QPE to model a wide range of downstream tasks, whether it involves classifying quantum phases of matter or predicting the entanglement entropy of quantum states. This adaptability is achieved simply by replacing the relevant inputs and outputs as needed. Rather than the *two-step* model (Wang et al., 2022) that uses the pretrained model to generate new measurement records conditioning on the physical variables and then predicts quantum properties based on classical shadow (Huang et al., 2020). LLM4QPE is an *end-to-end* task-agnostic pretrained model to provide property estimation for the quantum system.

**Quantum Data for Finetuning and Input Embeddings.** The dataset $\mathcal{D}_f = \{(\mathbf{R}_j, \mathbf{c}_j), \mathbf{p}_j\}_{i=j}^{N_f}$ are generated using the random seed different from the seed for generating $\mathcal{D}_p$. Then we split $\mathcal{D}_f$ to construct train/test dataset $\mathcal{D}_t / \mathcal{D}_e$. It is ensured that the sampled physical conditions for pretraining will not appear in finetuing, i.e. $\mathbf{c}_j \notin \{\mathbf{c}_i\}$ for $j \in \{1, \dots, N_f\}$. Note that the physical conditions for finetuning are sampled from the same distribution as the pretraining. The details about the data collection can be found in Appendix B. Unlike the pretraining where the input measurement records is a sentence-level vector $\boldsymbol{\sigma}_b \in \mathbb{Z}^L$, the input of fine-tuning becomes a batch of measurement records $\mathbf{X}_i \in \mathbb{Z}^{L \times K_f}$ where $K_f$ is the number of measurement strings. The reason for such change can be explained through both intuitive and rational perspectives. Intuitively, single measurement string cannot reflect the whole picture of the quantum system. Rationally, predicting the properties of the quantum system in classical computers generally requires exponential number of measurements with respect to the system size $L$ (Gebhart et al., 2023). Even though for some quantum systems with low entanglement, the number stills grows quasi-polynomially with $L$ (Huang et al., 2022). Accordingly, the input of the model is replaced with $\{(\mathbf{X}_j, \mathbf{c}_j), \mathbf{p}_j\}_{j=1}^{B_t}$ where the tuple $(\mathbf{X}_j, \mathbf{c}_j)$ is the input, $\mathbf{p}_j$

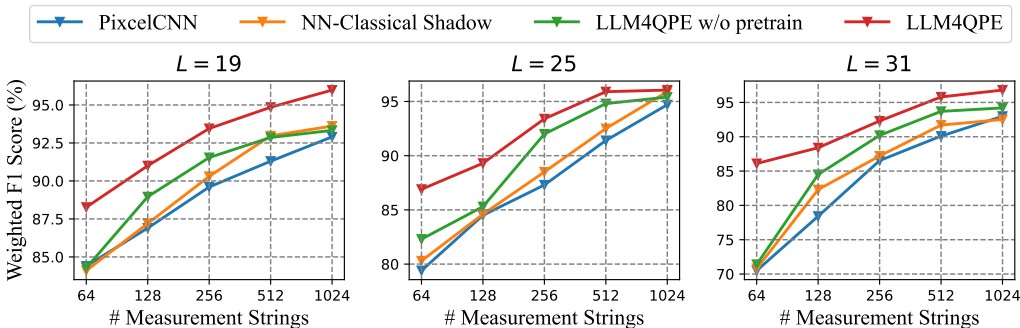

Figure 3: Comparison of weighted F1 score w.r.t. number of measurement strings on Rydberg atom model.

is the corresponding label and $B_t$ is the batch size used for supervised finetuning. The embedding is also distinct from that of pretraining. The learned token embeddings for the measurement string $\boldsymbol{\sigma}_i$ is not feasible for the batch-style records $\mathbf{X}_j$. To deal with it, a Long Short-Term Memory (LSTM) layer is attached in front of the decoder, as depicted in Fig. 1c. The LSTM layer converts the discrete measurement records $\mathbf{X}_j$ and outputs high-order embeddings $\mathbf{E}_{\text{rnn}} \in \mathbb{R}^{B_t \times L \times d}$. The additional embeddings including physical condition embeddings and positional embeddings are transferred from pretraining. The output embedding is the summation given as $\mathbf{E}_{\text{out}} = \mathbf{E}_{\text{rnn}} + \underbrace{\mathbf{E}_c + \mathbf{E}_p}_{\text{transferred}}$.

**Feature Aggregation and Output Projection.** The output of the $L$-layer transformer decoder is $\mathbf{H} \in \mathbb{R}^{B_t \times L \times d}$. For a specific downstream task, the decoder is initialized with the pretrained parameters and all the parameters are finetuned towards a supervised loss. To obtain the feature representation for each of the $B_t$ training samples, a feature aggregation layer is attached after the last multi-head attention layer. This layer converts the hidden feature $\mathbf{H}$ along the second axis and output $\mathbf{H}' \in \mathbb{R}^{B_t \times d}$. Finally, additional linear projection layer is employed to project the feature into $\mathbf{H}'' \in \mathbb{R}^{B_t \times P}$, along with a task-dependent activated function which is taken to be *tanh* for predicting the correlation function, since we have the prior that each element of the label $\mathbf{p}_j$ is in the range $[-1, 1]$ (See Appendix B for details). While the *log-softmax* is adopted for classifying quantum phases of matter.

**Learning Objective.** The properties estimation for the quantum system are treated as the supervised learning tasks. Tow types of tasks are considered in this paper, including classifying quantum phases of matter and predicting correlation function. The former belongs to the regression task, while the latter can be regarded as a classification task. For each supervised task, we maintain a consistent architecture within LLM4QPE. We seamlessly integrate task-specific inputs and ground-truth labels into LLM4QPE and proceed to finetune all model's parameters in an end-to-end manner. Given that the training samples are $\{(\mathbf{X}_j, \mathbf{c}_j), \mathbf{p}_j\}_{j=1}^{B_t}$ where $B_t$ is the batch size. For classifying quantum phases of matter, $\mathbf{p}_j$ is the one-hot label. We minimize the observed data negative log-likelihood which yields a supervised loss for classification (with $P$ classes):

$$\mathcal{L}_{\text{sup}} = -\frac{1}{B_t} \sum_{j \in \{1, \dots, N_t\}} \sum_{u=1}^{P} \mathbb{I}\left[\mathbf{p}_{j,u} = 1\right] \log\left(f_{\boldsymbol{\theta}}\left(\mathbf{X}_j, \mathbf{c}_j\right)_u\right), \tag{4}$$

where $\mathbb{I}[\cdot]$ is an indicator function, $N_t$ is the size of training dataset and $f_{\boldsymbol{\theta}}(\cdot)$ denotes the prediction of the model with parameters $\boldsymbol{\theta}$ to be optimized. For predicting the correlation, $\mathbf{p}_j$ is the continuous valued label. We adopt the Root Mean Square Error (RMSE) loss:

$$\mathcal{L}_{\text{sup}} = \sqrt{\tilde{\mathcal{L}}_{\text{sup}}}, \quad \tilde{\mathcal{L}}_{\text{sup}} = \frac{1}{B_t} \sum_{j \in \{1, \dots, N_t\}} \sum_{u=1}^{P} \left(f_{\boldsymbol{\theta}}\left(\mathbf{X}_j, \mathbf{c}_j\right)_u - \mathbf{p}_{j,u}\right)^2. \tag{5}$$

Detailed description of task-specific finetuning can be found in the experiment section.

## 4 EXPERIMENTS

In this section, we present the finetuning results on two quantum property estimation tasks including classifying quantum phases of matter and predicting correlation function. Two families of quan-

Table 1: Classification accuracy of quantum phases of matter on the Rydberg atom model with varied system size $L$ and varied training size $N_t$, where $K_f$ is fixed to be 1024. The best results are highlighted in **bold**.

| Method | $L = 19$ | | | $L = 25$ | | | $L = 31$ | | |
|---|---|---|---|---|---|---|---|---|---|
| | $N_t = 25$ | $N_t = 64$ | $N_t = 100$ | $N_t = 25$ | $N_t = 64$ | $N_t = 100$ | $N_t = 25$ | $N_t = 64$ | $N_t = 100$ |
| RBF Kernel | 91.75 | 92.29 | 93.25 | 88.43 | 92.27 | 94.2 | **88.32** | 90.79 | 92.75 |
| NTK | 92.12 | 92.58 | 93.79 | 89.17 | 94.14 | 95.39 | 86.99 | 92.03 | 92.71 |
| PixelCNN | 92.18 | 92.79 | 92.98 | 88.91 | 91.59 | 94.73 | 85.29 | 92.21 | 92.98 |
| NN-shadow | 91.73 | 92.64 | 93.61 | 90.57 | 91.32 | 95.91 | 86.38 | 91.79 | 92.51 |
| LLM4QPE | **94.14** | **93.38** | **95.95** | **93.95** | **96.51** | **96.05** | 87.95 | **94.95** | **96.67** |
| LLM4QPE w/o pretrain | 93.80 | 92.89 | 93.35 | 90.85 | 95.35 | 95.27 | 87.45 | 92.77 | 94.32 |

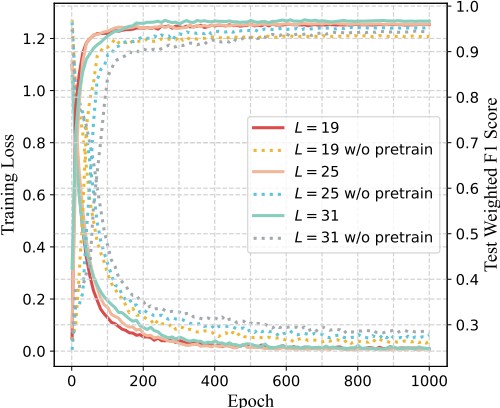

Figure 4: The evolution of training loss and test weighted F1 score with increasing training epochs where $N_t = 100$ and $K_f = 1024$.

Figure 5: The required number of epochs for each respective model to attain 90% of its peak weighted F1 score where $N_t = 100$.

tum models are considered – the Rydberg atom model (Bernien et al., 2017) and the anisotropic Heisenberg model (Kranzl et al., 2023).

As baseline methods, we basically consider the classical shadow (Huang et al., 2020) – a learning-free protocol for constructing the representation of an unknown quantum state. Besides, we compare with some kernel methods including Radial Basis Function (RBF) Kernel (Huang et al., 2022) and Neural Tangent Kernel (NTK) (Huang et al., 2022). We further consider some advanced deep learning based methods, such as PixelCNN (Sharir et al., 2020) and a classical shadow based generative model (NN-shadow) (Wang et al., 2022) for comparison.

### 4.1 CLASSIFYING QUANTUM PHASES OF MATTER ON RYDBERG ATOM MODEL

We first consider the Rydberg atom model with different system size $L \in \{19, 25, 31\}$. We pretrain LLM4QPE for different system sizes separately with a fixed number of sampled physical conditions $N_p = 100$. Each physical condition variable $\mathbf{c}_i$ is a 4-dimensional vector denoted as $[L_i, \Delta_i, \Omega_i, R_0/a_i]^\top$ where $\Delta$ is the detuning of a laser, $\Omega$ is the Rabi frequency and $R_0/a$ is the interaction range. The values of these four variables can be obtained directly when initializing the (simulated) quantum experiments. For each physical condition we generate $K_f$ measurement strings based on computational basis measurement operators, such that the total number of possible measurement outcomes is $M = 2$. Then LLM4QPE is pretrained with dataset $\mathcal{D}_p$. The pretrained parameters are transferred to finetune the model using $\mathcal{D}_t$, where the number of sampled physical conditions $N_t \in \{25, 64, 100\}$ and the number of measurement strings $K_f \in \{64, 128, 256, 512, 1024\}$. We fix the size of $\mathcal{D}_e$ for evaluation to be $N_e = 10000$. Following (Bernien et al., 2017), we consider three categories of quantum phase, i.e., Disorder, $Z_2$, $Z_3$ to establish the label $\mathbf{p}_j$, which is a 3-dimensional one-hot vector. More details about the data generation can be found in Appendix B.

We also take evaluation without pretaining the LLM4QPE: all the parameters are initialized randomly in a uniform distribution $[-1, 1]$. We use *accuracy* and *weighted F1 score* as metrics for 3-class classification for evaluation of our models and baselines. The results are listed in Tab. 1 and LLM4QPE achieves the best mean accuracy except for one setting $L = 31$ with $N_t = 25$.

Table 2: RMSE of predicting the correlation on the anisotropic Heisenberg model with varied system size $L$ and training size $N_t$. $K_f$ is fixed to 64. The best results are in **bold**.

| Method | $L = 8$ | | | $L = 10$ | | | $L = 12$ | | |
|---|---|---|---|---|---|---|---|---|---|
| | $N_t = 20$ | $N_t = 50$ | $N_t = 90$ | $N_t = 20$ | $N_t = 50$ | $N_t = 90$ | $N_t = 20$ | $N_t = 50$ | $N_t = 90$ |
| Classical Shadow | 0.2015 | 0.1954 | 0.1967 | 0.2015 | 0.1997 | 0.2015 | 0.1991 | 0.2064 | 0.2117 |
| RBF Kernel | 0.2085 | 0.2077 | 0.2081 | 0.2104 | 0.2131 | 0.2079 | 0.2039 | 0.1931 | 0.2157 |
| NTK | 0.2062 | 0.2064 | 0.2052 | 0.2095 | 0.2085 | 0.2097 | 0.2141 | 0.1922 | 0.2105 |
| PixelCNN | 0.2257±0.015 | 0.2357±0.019 | 0.2239±0.024 | 0.2393 | 0.2289±0.023 | 0.2108±0.024 | 0.2390±0.024 | 0.2297±0.035 | 0.2267±0.038 |
| NN-shadow | 0.2069±0.022 | 0.2098±0.015 | 0.2057±0.012 | 0.2078±0.017 | 0.2054±0.017 | 0.1959±0.013 | 0.2037±0.029 | 0.2021±0.019 | 0.2102±0.026 |
| LLM4QPE | **0.1761±0.032** | **0.1612±0.022** | **0.1697±0.025** | **0.1986±0.011** | **0.1949±0.012** | **0.1893±0.023** | **0.1989±0.023** | **0.1787±0.021** | **0.1769±0.015** |
| LLM4QPE w/o pretrain | 0.2043±0.027 | 0.2057±0.036 | 0.1949±0.027 | 0.2179±0.015 | 0.1984±0.013 | 0.1981±0.025 | 0.2040±0.028 | 0.2097±0.031 | 0.2026±0.027 |

Table 3: Ablation study results on condition embedding and LSTM embedding. We consider $N_t = 64$ with $K_f = 1024$ for the Rydberg model, and $N_t = 50$ with $K_f = 64$ for the Heisenberg model.

| Rydberg | $L = 19$ | $L = 25$ | $L = 31$ | Heisenberg | $L = 8$ | $L = 10$ | $L = 12$ |
|---|---|---|---|---|---|---|---|
| original | **93.38** | **96.51** | **94.95** | original | **0.1612** | **0.1949** | **0.1787** |
| w/o cond. embed. | 93.29 | 95.96 | 93.52 | w/o cond. embed. | 0.1906 | 0.2095 | 0.1981 |
| w/o LSTM embed. | 90.75 | 92.18 | 89.65 | w/o LSTM embed. | 0.1929 | 0.1997 | 0.1904 |

Fig. 3 shows the performance on varied $K_f$. LLM4QPE achieves the best weighted F1 score across all systems and in particular, outperforms by a large margin when $K_f = 64$. The results indicate that pretrained LLM4QPE can handle the input when a few number of measurement records are available, which is greatly instrumental due to the expensive and time-consuming (simulated) quantum experiments. We further plot the training dynamics of LLM4QPE with and without pretraining throughout the training epochs in Fig. 4. The curves indicate that the pretraining enables much faster convergence of supervised loss and achieves better finetuning accuracy. Meanwhile, the required number of epoch for the model to attain 90% of its peak weighted F1 score is provided in Fig. 5. It reflectw that within the same system size $L$, the pretrained LLM4QPE converges faster than the non-pretrained version, with a lower training error and a higher test weighted F1 score.

### 4.2 PREDICTING CORRELATION FUNCTION ON ANISOTROPIC HEISENBERG MODEL

Next we consider a regression task - predicting correlation on the anisotropic Heisenberg model. This quantum model inherits the long-range interactions between every two quantum sites, leading to a complex dynamics which is hard to be simulated by classical computers (Orús, 2019). We restrict the system size $L \in \{8, 10, 12\}$ due to memory limitations. The ground states of quantum systems with different physical conditions are calculated by eigenvalue decomposition. For each physical condition we generate $K_f$ measurement strings based on Pauli-6 measurement operators such that $M = 6$. Then we pretrain the LLM4QPE for different system sizes independently with training size $N_p = 100$.

For model's finetuning, we vary the number of generated training samples $N_t \in \{20, 50, 90\}$ and fix the measurement strings $K_f = 64$. The dataset used for evaluation is generated with $N_e = 200$. To obtain the ground-truth labels, We calculate true values of the two-body correlation functions and collect them as the supervised labels, which is an $L \times L$ continuous-valued matrix where each entry is in the range $[-1, 1]$. The RMSE results is reported in Tab. 2. LLM4QPE outperforms baselines in all settings. The learning-based models baselines often fail to surpass the predictive accuracy of learning-free classical shadow. While our pretrained LLM4QPE stands out by a remarkable margin.

Finally, we study the effects of condition embedding and the LSTM embedding on both Rydberg atom model and anisotropic Heisenberg model. Note that we replace the LSTM with a fully connected layer with same input/output dimension. The results are given in Tab. 3, where the results consistently show that both embedding techniques contribute to some positive effects and suggest that these two techniques can both help to leverage useful information from input quantum data.

## 5 CONCLUSION AND OUTLOOK

This paper proposes a task-agnostic unsupervised pretraining approach for estimation of the properties of the quantum systems via quantum datasets. The core of our approach is a transformer encoder enabling to learn useful hidden information in a fully unsupervised pretraining procedure. The pretrained parameters can be transferred to solving downstream tasks, leading to more effective classifying quantum phases and predicting correlation function on a resource-limited device given limited measurement information.

ETHICS STATEMENT

This paper proposes a novel approach for estimating the properties of quantum systems inspired by LLMs. The authors acknowledge the potential ethical implications of this research, such as the misuse of quantum data, the bias or error in the estimation results, and the impact on the development of quantum technologies. The authors have followed the best practices for data collection, model design, and evaluation, and have disclosed the sources of funding and the conflicts of interest. The authors also adhere to the principles of research integrity and comply with the relevant laws and regulations. The authors hope that this research will contribute to the advancement of quantum science and benefit the future research.

REPRODUCIBILITY STATEMENT

The generated quantum data of the Rydberg atom model and the anisotropic Heisenberg model is available at https://github.com/abel1231/qpe-data. The code to train the model and analyze the experimental results is available from the first author on reasonable request.

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

# A   RELATED WORK

## A.1   LEARNING-FREE METHODS FOR QPE

Estimating the properties of the quantum system is a long-standing problem in quantum physics (D'Ariano et al., 2003). The main challenge is that the complexity of describing the quantum system using classical computers typically scales exponentially with respect to the system size (Nielsen & Chuang, 2010). Even though, in fact, the quantum systems studied in physical experiments generally can be described by a limited number of physical variables. This restriction leads to the studied quantum systems occupy only a small part of the exponentially large Hilbert space (Carrasquilla et al., 2019), such that they can be characterized by some classical methods within an acceptable error.

Traditional algorithms including the QMC (Ceperley & Alder, 1986) and DFT (Hohenberg & Kohn, 1964) has made success for investigating the electronic structure (or nuclear structure), principally the ground state of many-body systems, such as atoms, molecules, and the condensed phases (Gubernatis et al., 2016). However, these methods have scalability issues and are difficult to be used to deal with large-scale quantum many body problems. An alternative is a class of TNs methods (Orús, 2019) based on variational method and shows unprecedented performance in analyzing the characteristics of ground state. These methods including Matrix Product State (MPS) (Perez-Garcia et al., 2006) and Projected Entangled Pair States (PEPS) (Corboz, 2016). This class of methods approximates the wave function by decomposition of the high-order wave functions into multiple low-rank tensors. It is then possible to analyze properties of the quantum state by taking algebra operations on the wave function. Recently, the classical shadow protocol (Huang et al., 2020) suggests to use random measurements to characterize the quantum properties. Classical shadow has facilitated applications such as direct fidelity estimation (Struchalin et al., 2021) and state function prediction (Zhang et al., 2021).

## A.2   LEARNING-BASED METHODS FOR QPE

With the continuous development of machine learning technologies, neural network based methods have emerged to tackle the QPE problems. These methods can be categorized into two classes according to the purpose. The methods (Carleo & Troyer, 2017; Gao & Duan, 2017; Torlai et al., 2018; Schütt et al., 2019; Hibat-Allah et al., 2020; Zhang & Di Ventra, 2023) of the first class are called Neural Network Quantum State (NNQS), which replace the tensor used in TNs with a neural network as a parametric function approximator of quantum many-body wave functions. The parameterized wave function is updated by minimizing the expectation values of relevant observable estimators, based on either density matrix renormalization group (DMRG) algorithm (White, 1992) or variational Monte Carlo (VMC) (McMillan, 1965). Afterwards the interested properties can be analyzed by preforming algebra operations on the wave function. Another line of research (Gilmer et al., 2017; Kawai & Nakagawa, 2020; Xiao et al., 2022) is known as Neural Network Quantum Property Estimation (NNQPE). NNQPE directly optimizes the parameters towards a specific learning objective which represents a certain property of quantum systems such as the quantum phase.

For both NNQS and NNQPE, different neural network ansatz corresponds to solve quantum many-body problems with different physical structures. Examples include restricted Boltzmann machine (RBM) (Carleo & Troyer, 2017), recurrent neural networks (RNNs) (Carrasquilla et al., 2019), convolutional neural networks (CNNs) (Wu et al., 2019; Sharir et al., 2020; Wu et al., 2023), and transformers (Cha et al., 2021; Wang et al., 2022; Zhang & Di Ventra, 2023; Du et al., 2023).

Our work is closely related to NNQPE. While ours employs a unsupervised pretraining to extract the hidden information of the quantum systems govern by different parameters. We find empirically that this scheme can make the model perform better under a limited number of copies of quantum states and measurements. The recent work proposed by Zhu et al. (2022) implements a similar pretraining strategy for learning of quantum states, whereas our approach differs from it by avoiding assumptions about knowing the prior frequency about the measurement strings.

# B   DETAILS OF THE QUANTUM DATASET GENERATION

A quantum dataset is a collection of data that describes quantum systems and their evolution. The collection of quantum data must take into account the following factors:    1) the method of data collection must be feasible on quantum devices and not contradict the disciplines of quantum mechanics; 2) the process of data collection is completely automated and does not require experienced experts to organize and label it and 3) the data must be structured and can be stored on resource-limited classical devices, thus can be easy to be processed by the machine learning techniques without further post-processing. The quantum dataset we established satisfies these three points. It is also worth mentioning that our model can be used as a centralized infrastructure to process all these data uniformly, thanks to the unsupervised pretraining design of the model.

In this paper, we conduct simulated experiments to generate the quantum dataset in classical computers. For the anisotropic Heisenberg model, quantum measurement is performed using the Pauli-6 measurement operators such that $M = 6$, whereas computational basis measurement operators are employed for the Rydberg atom model leading to $M = 2$. Assume that variables $\mathbf{c}_i$ describing the physical condition lives in a finite continuous space $\mathbb{F}$ within the physical restriction. Let $\mathcal{D}_p = \{\mathbf{R}_i, \mathbf{c}_i\}_{i=1}^{N_p}$ denote the quantum dataset used for pre-training and $\mathcal{D}_f = \{(\mathbf{R}_i, \mathbf{c}_i), \mathbf{p}_i\}_{i=1}^{N_f}$ for fine-tuning, where $|\mathcal{D}_p| = N_p$ and $|\mathcal{D}_f| = N_f$. For pre-training the model, we first uniformly sample a number of points $\{\mathbf{c}_i | \mathbf{c}_i \in \mathbb{F}\}_{i=1}^{N_p}$. Afterwards we conduct simulated experiments for each $\mathbf{c}_i$ and collect the corresponding measurement records. The system property $\mathbf{p}_i$ is not needed since the pre-training phase is fully unsupervised. While for fine-tuning, we initialized the experiments with another random seed and sample $N_f$ physical conditions also within space $\mathbb{F}$, resulting in $\{\mathbf{c}_j | \mathbf{c}_j \in \mathbb{F}\}_{j=1}^{N_f}$. Note that We also collect the measurement records for each $\mathbf{c}_j$. The difference part is that we additionally calculate the system property $\mathbf{p}_j$ and use it as supervised labels. We further split the $\mathcal{D}_f$ into $\mathcal{D}_t$ and $\mathcal{D}_e$ for training and evaluation respectively with varied separation ratio. Details of the hyper-parameters and the experimental configurations of the dataset generation are discussed below.

## B.1   RYDBERG ATOM MODEL

Rydberg atom model is a programmable quantum simulators capable of preparing interacting qubit systems (Bernien et al., 2017). Such quantum model can be effectively described as a two-level quantum system consisting the ground state $|g\rangle$ $(|0\rangle)$ and the Rydberg state $|r\rangle(|1\rangle)$. The quantum dynamics of this model is governed by the Hamiltonian

$$H_{\text{Rydberg}} = \sum_i \frac{\Omega}{2}\sigma_x^i - \sum_i \Delta n_i + \sum_{i<j} \frac{V_0}{|\vec{x}_i - \vec{x}_j|} n_i n_j \tag{6}$$

where $\sigma_x$ is the Pauli-X matrix, $\Omega$ is the Rabi frequency, $\Delta$ is the detuning of a laser, $V_0$ is the Rydberg interaction constant, $i, j$ is the Rydberg interaction constant and $\vec{x}_i$ is the position vector of the site $i$. $n_i = |r_i\rangle\langle r_i|$ is the occupation number operator at site $i$, and $\sigma_x^i = |g_i\rangle\langle r_i| + |r_i\rangle\langle g_i|$ describes the coupling between the ground state $|g_i\rangle$ and the Rydberg state $|r_i\rangle$ at position $i$.

We follow the recent work in (Wang et al., 2022) to generate the quantum dataset. We refer the readers to their paper for details. Here we briefly introduce the main procedures. We consider the Rydberg atom model with system size $L \in \{19, 25, 31\}$. We fix the interaction constant $V_0 = 862690 \times 2\pi$ MHz $\mu m^6$ and vary the value of $\Omega \in [0, 5]$ and $\Delta \in [-10, 15]$ to get different physical conditions $\mathbf{c}$, where $\mathbf{c}$ is a 4-dimensional vector in the form $[L, \Delta, \Omega, R_0/a]$, where $R_0/a$ denote the interaction range with $R_0 = (V_0/\Omega)^{1/6}$. Then the approximate ground state for diffident physical condition is prepared by the tool `Bloqade.jl` (blo, 2023). This tool can also output the measurement strings and the true phase of each physical condition. The measurement operators are chosen to be the computational basis $\{|0\rangle\langle 0|, |1\rangle\langle 1|\}$ for the quantum measurement, such that the total number of the possible outcomes is $M = 2$. In this paper, three different phases are considered including the Disordered phase, $Z_2$ Ordered phase and $Z_3$ Ordered phase. We sample $N_p = 100$ physical conditions with $K_p = 1024$ measurement strings for pre-training, and $N_t \in \{25, 64, 100\}$ physical conditions with $K_f \in \{64, 128, 256, 512, 1024\}$ for fine-tuning. The number of physical conditions for evaluation is fixed to be $N_e = 10000$. The supervised labels for fine-tuning are one-hot encoded vectors of the true phases such that the dimension (number of classes) of $\mathbf{p}$ is 3. Note that it is ensured that the sampled physical conditions for pre-training will not appear in fine-tuning.

## B.2 Anisotropic Heisenberg Model

Exploring the effects of these long-range interactions of the quantum system is essential for understanding the quantum mechanics (Bermúdez et al., 2017). In this paper, we consider the recent progress for the long-range interactions with the experimentally realized power-law exponent of the anisotropic Heisenberg model (Kranzl et al., 2023). The dynamics of the anisotropic Heisenberg model is determined by the Hamiltonian

$$H_{\text{Heisenberg}} = \frac{1}{3} \sum_{i<j} J_{ij}(\sigma_x^i \sigma_x^j + \sigma_y^i \sigma_y^j + h\sigma_z^i \sigma_z^j), \tag{7}$$

where $\sigma_{x,y,z}^i$ is the Pauli matrix operated on the $i$-th site, $h$ determines the Ising interactions between the magnons, and $J_{ij}$ is the long-range interaction strength satisfying $J_{ij} = J/|i - j|^\alpha$. We follow the configuration of (Kranzl et al., 2023) to geenrate the quantum dataset. The values of $h$ and $J$ are fixed with 1 and 369 rad/s, and we vary the value of $\alpha \in (1, 2]$ uniformly. It is extremely hard to characterize the quantum system with long-range interactions using the existing computing techniques. Thus we restrict the system size $L \in \{8, 10, 12\}$. For all the systems we consider the number of measurement strings used for pre-training as $K_p = 1024$ and fix the number of sampled physical conditions as $N_p = 100$. For model's finetuning, we vary the number of generated training samples $N_t \in \{20, 50, 90\}$ and fix the measurement strings $K_f = 64$. The physical condition **c** is defined as a vector whose dimension $C = L^2$, in which each element is the coupling strength $J_{ij}$ for $i, j \in \{1, \ldots, L\}$. The problem of finding the ground state is viewed as the eigenvalue decomposition problem and we obtain the ground state for each sampled physical condition by the `scipy` (Virtanen et al., 2020) built-in functions. The measurement records and the true values of the two-body correlation function and the entanglement entropy are obtained using the `pennylane` (Bergholm et al., 2018) toolbox. We consider the Pauli-6 POVM measurement operators with $M = 6$ outcomes, which are given as $M_{\text{Pauli-6}} = \left\{ \frac{1}{3} \times |0\rangle\langle0|, \frac{1}{3} \times |1\rangle\langle1|, \frac{1}{3} \times |+\rangle\langle+|, \frac{1}{3} \times |-\rangle\langle-|, \frac{1}{3} \times |r\rangle\langle r|, \frac{1}{3} \times |l\rangle\langle l| \right\}$, and $\{|0\rangle, |1\rangle\}, \{|+\rangle, |-\rangle\}, \{|r\rangle, |l\rangle\}$ stand for the eigenbasis of the Pauli operators $\sigma_z, \sigma_x$, and $\sigma_y$, respectively. For the task of predicting the correlation matrix, the ground-truth label is a $L \times L$ matrix and each element of the matrix is the expectation value of the observable

$$O_{ij} = \frac{1}{3} \left( \sigma_x^i \sigma_x^j + \sigma_y^i \sigma_y^j + \sigma_z^i \sigma_z^j \right). \tag{8}$$

Thus each element can be written as $\text{tr}(\rho O_{ij})$ in the range $[-1, 1]$, where $\rho$ is the density matrix of the ground state for each sampled physical condition. We flatten the correlation function matrix to be the $L^2$-dimensional continuous-valued vector and treat it as the supervised label for fine-tuning. While for the task of predicting the entanglement entropy, the label is a real number which can be calculated as $-\log(\text{tr}(\rho_A^2))$, where $A$ is the left-half subsystem with system size $L/2$ of the $L$-qubit quantum system.

## C Poof of the Normalized Output Distribution

In the main text, we claim that the output (classical) distribution satisfies

$$\sum_{\sigma_1=1}^{M} \cdots \sum_{\sigma_L=1}^{M} p(\sigma_1, \ldots, \sigma_L) = 1, \tag{9}$$

as long as the last linear projection layer uses the *softmax* activated function. The proof is given below.

The *softmax* activated function is performed on the model's output, which is the product of conditional probabilities $p(\sigma_1, \ldots, \sigma_L) = \prod_{i=1}^{L} p(\sigma_i | \sigma_{i-1}, \ldots, \sigma_1)$. It is easy to check the claim holds for $L = 1$. Given that the claim also holds for $L = k$. For $L = k + 1$, the following equation then be hold:

$$\sum_{\sigma_i=1}^{M} p(\sigma_i | \sigma_{i-1}, \ldots, \sigma_1) = 1. \tag{10}$$

Table 4: The RMSE of predicting the second-order Rényi entanglement entropy for the anisotropic Heisenberg model. We sample $N_p = 100$ physical conditions with $K_p = 1024$ measurement strings for pre-training.

| Method | $L = 8$ | | | | | $L = 10$ | | | | | $L = 12$ | | | | |
|---|---|---|---|---|---|---|---|---|---|---|---|---|---|---|---|
| | $K_f = 64$ | $K_f = 128$ | $K_f = 256$ | $K_f = 512$ | $K_f = 1024$ | $K_f = 64$ | $K_f = 128$ | $K_f = 256$ | $K_f = 512$ | $K_f = 1024$ | $K_f = 64$ | $K_f = 128$ | $K_f = 256$ | $K_f = 512$ | $K_f = 1024$ |
| Classical Shadow | 1.58282 | 1.56688 | 1.50989 | 1.40270 | 1.22974 | 1.72379 | 1.71451 | 1.73135 | 1.72740 | 1.68556 | 2.89481 | 2.90874 | 2.91391 | 2.90773 | 2.89722 |
| RBF Kernel | 0.07322 | 0.07160 | 0.07670 | 0.07692 | 0.07706 | 0.02539 | 0.02257 | 0.02242 | 0.02002 | 0.01983 | 0.08710 | 0.08242 | 0.08104 | 0.07081 | 0.07032 |
| NTK | 0.07117 | 0.06799 | 0.08834 | 0.08708 | 0.08690 | 0.02497 | 0.02221 | 0.02129 | 0.01996 | 0.01947 | 0.08432 | 0.08249 | 0.08071 | 0.07998 | 0.07381 |
| PixelCNN | 0.07198 | 0.07091 | 0.06849 | 0.06687 | 0.06784 | 0.01907 | 0.01892 | 0.01948 | 0.01952 | 0.02089 | 0.07406 | 0.07145 | 0.07107 | 0.06895 | 0.06677 |
| NN-shadow | 0.06860 | 0.06415 | 0.06403 | 0.06315 | 0.06221 | 0.01844 | 0.01747 | 0.01664 | 0.01662 | 0.01657 | 0.07261 | 0.06858 | 0.06573 | 0.06156 | 0.05924 |
| LLM4QPE | **0.06302** | **0.06141** | **0.06104** | **0.05998** | 0.06072 | **0.01698** | **0.01623** | **0.01534** | **0.01517** | **0.01520** | **0.05861** | **0.05812** | **0.05648** | **0.05623** | **0.05597** |
| LLM4QPE w/o Pretrain | 0.06649 | 0.06295 | 0.06228 | 0.06071 | **0.06034** | 0.01711 | 0.01662 | 0.01696 | 0.01655 | 0.01532 | 0.06624 | 0.06542 | 0.06381 | 0.06042 | 0.05931 |

Such that

$$
\begin{aligned}
&\sum_{\sigma_1=1}^{M} \cdots \sum_{\sigma_{k+1}=1}^{M} p(\sigma_1, \ldots, \sigma_{k+1}) \\
&= \sum_{\sigma_1=1}^{M} \cdots \sum_{\sigma_{k+1}=1}^{M} |\boldsymbol{\Psi}(\sigma_1, \ldots, \sigma_{k+1})|^2 \\
&= \sum_{\sigma_1=1}^{M} \cdots \sum_{\sigma_{k+1}=1}^{M} \prod_{i=1}^{k+1} |\boldsymbol{\Psi}(\sigma_i|\sigma_{i-1}, \ldots, \sigma_1)|^2 \\
&= \sum_{\sigma_1=1}^{M} \cdots \sum_{\sigma_k=1}^{M} \left( \prod_{i=1}^{k} |\boldsymbol{\Psi}(\sigma_i|\sigma_{i-1}, \ldots, \sigma_1)|^2 \right) \sum_{\sigma_{k+1}=1}^{M} |\boldsymbol{\Psi}(\sigma_{k+1}|\sigma_k, \ldots, \sigma_1)|^2 \\
&= \sum_{\sigma_1=1}^{M} \cdots \sum_{\sigma_k=1}^{M} |\boldsymbol{\Psi}(\sigma_1, \ldots, \sigma_k)|^2 \\
&= 1
\end{aligned} \tag{11}
$$

The proof then complete.

# D  ADDITIONAL NUMERICAL RESULTS

## D.1  RESULTS OF PREDICTING THE ENTANGLEMENT ENTROPY

We take an additional downstream task: predicting the second-order Rényi entanglement entropy $-\log(\mathrm{tr}(\rho_A^2))$ for the anisotropic Heisenberg model, where $A$ is the left-half subsystem with system size $L/2$ of the $L$-qubit quantum system. The number of training size is set to be $N_t = 90$ and the predicted RMSE results are given in Tab. 4. It can be observed that pre-training remains effective for predicting the entanglement entropy of the anisotropic Heisenberg model.

## D.2  MODEL SENSITIVITY TO THE NUMBER OF MEASUREMENTS

In Sec. 4, we study the relationship between the number of measurements and the classification accuracy of quantum phase of matters on Rydberg atom model. It is empirically evident in Fig. 3 that achieving linear growth in classification accuracy requires an exponential increase in the number of measurements per training example. Beyond the scaling related to number of measurements, we dive into further research on the scaling relationship between accuracy and the size of the training set (i.e., the number of sampled physical conditions which determine the dynamics of the quantum systems). We constrain the number of measurement per example to 256 (since we find that a large value makes the accuracy reach saturation) and the results on the 31-qubit system are listed in the Tab. 5. The results show that the accuracy approximately exhibits linear growth w.r.t. training size. This finding is consistent with theoretical results presented in (Huang et al., 2022; Lewis et al., 2024), which demonstrate that there exists a polynomial scaling relationship between model performance and the size of training dataset.

Table 5: Classification accuracy of quantum phases of matter on the Rydberg atom model with varied training size $N_t$, where $L = 31$ and $K_f = 256$. The results are averaged over 3 runs with different random seeds.

|  | $N_t = 20$ | $N_t = 40$ | $N_t = 60$ | $N_t = 80$ |
|---|---|---|---|---|
| LLM4QPE | 82.05 | 87.24 | 89.16 | 90.63 |
| LLM4QPE w/o pretrain | 79.17 | 81.78 | 85.96 | 88.47 |

Table 6: Classification accuracy of quantum phases of matter on the 31-qubit Rydberg atom model. The pre-trained parameters are transferred from the model trained on smaller system size. The training size is set to be $N_t = 100$, and the number of measurements $K_f = 1024$.

| | |
|---|---|
| LLM4QPE (pre-trained on 19-qubit system) | 95.74 |
| LLM4QPE (pre-trained on 25-qubit system) | 96.13 |
| LLM4QPE (pre-trained on 31-qubit system) | 96.67 |
| LLM4QPE w/o pre-train | 94.32 |

Table 7: Classification accuracy of quantum phases of matter on the 19-qubit Rydberg atom model. The training size is set to be $N_t = 100$, and the number of measurements $K_f = 1024$.

|  | no OOD | OOD |
|---|---|---|
| LLM4QPE | 95.95 | 84.82 |
| LLM4QPE w/o pre-train | 93.35 | 94.23 |

### D.3 FINE TUNING THE MODEL WITH OUT-OF-DISTRIBUTION DATASET

In this section, we consider fine tuning the LLM4QPE with out-of-distribution (OOD) dataset, which means the dataset used for fine-tuning and the dataset used for pre-training come from different distributions.

Here, we consider two different configurations to make the fine-tuning dataset out-of-distribution from the pre-training one: the first is to re-generate the fine-tuning data by modifying the physical variables and the second is to fine tune the model based on the parameters transferred from the model pretrained on fewer qubits. In the following, we consider the Rydberg atom model.

First, we take the evaluation that fine-tuning the model on 31-qubit system by using he parameters pre-trained on 19 and 25-qubit system. Note that the number of qubits is also a physical variable and we want to see if model parameters trained on small-scale systems could transfer and help model characterize larger-scale systems. The results are listed in Tab. 6. It is evident that pre-trained parameters transferred from small-scale systems is also useful for large-scale systems.

Second, we modify the detuning of a laser from $[-10, 15]$ (which is exactly used in the paper) to $[-20, -10] \cup [15, 25]$ to generate OOD fine-tuning dataset, on Rydberg atom model with 19 qubits. The classification accracy are listed in Tab. 7. The pre-trained one fails to perform better than the LLM4QPE w/o pre-train. The main reason is that the modified detuning values have driven the quantum evolution into a very different dynamics and the pre-trained model learns less knowledge about it. Whether pre-training of LLM4QPE remains beneficial for OOD quantum datasets in other settings remains an open question, and will be further explored in our future work.

## E LIMITATIONS

In this study, we concentrate on the classification of quantum phases of matter and the prediction of correlation functions for the Rydberg atom model and the anisotropic Heisenberg model, respectively. While the LLM4QPE model offers flexibility for addressing various quantum many-body challenges, such as reconstructing the density matrix. Our focus here is primarily on pretraining the model with a fixed number of measurement strings. The impact of varying the number of measurement strings on the model's performance presents a fascinating area for exploration. Additionally,

the LLM4QPE model is characterized by a relatively small parameter count (tens of thousands of parameters) when compared to the significantly larger parameter sets of large language models. Due to the constraints imposed by the model's size, our pretraining efforts are confined to quantum systems govern by Hamiltonians from the same family. Looking forward, there is an anticipation to develop a more robust model, enriched with a greater number of parameters, through learning on datasets generated from diverse families of quantum systems.

