# OpenReview forum: "Towards LLM4QPE: Unsupervised Pretraining of Quantum Property Estimation and A Benchmark"
_ICLR.cc/2024/Conference — ICLR 2024 spotlight_

### Official Review · Reviewer_TiJ4 · 2023-10-29

**Soundness:** 3 good
**Presentation:** 4 excellent
**Contribution:** 3 good
**Rating:** 8
**Confidence:** 4

**Summary:**

This manuscript introduces a framework named QTAPE designed to address the challenge of quantum property estimation. The authors utilize an neural network-based model to leverage different types of quantum data to make the estimation.  Additionally, they showcase the model's effectiveness by deploying it to estimate properties in various many-body systems.

**Strengths:**

- This manuscript employs a framework composed of unsupervised pre-training and supervised fine-tuning phase for estimating the properties.

- The proposed method demonstrates versatility, as evidenced by its effectiveness across various types of tasks, as illustrated by the numerical experiments.

- The proposed model exhibits satisfactory performance even in large-scale quantum systems.

- The presentation of the paper is excellent.

**Weaknesses:**

- While the authors emphasize the unsupervised pre-training as a key novelty of their proposed method, I just feel that the approach they employ in the pre-training phase, involving the regeneration of measurement results, follows a similar logic to that found in [1]. In [1], the model generates measurement results from incomplete measurement results. Although the proposed model here incorporates additional input information, such as physical conditions, I still believe the authors should cite this previous work.
- Furthermore, the fine-tuning phase of the model shares similarities to the framework described in [2]. Despite the results demonstrating the superiority of the proposed method in Table 1, it would be valuable for the authors to engage in a more in-depth discussion regarding the distinctions between their model and this prior work. Given that Q-TAPE without pretraining also outperforms NN-Classical Shadow, it would be appreciated if the authors could provide further analysis or intuitive explanations for the results to highlight the novelty of the proposed model.

[1] Zhu, Y., Wu, Y. D., Bai, G., Wang, D. S., Wang, Y., & Chiribella, G. (2022). Flexible learning of quantum states with generative query neural networks. Nature Communications, 13(1), 6222.

[2] Wang, H., Weber, M., Izaac, J., & Lin, C. Y. Y. (2022). Predicting Properties of Quantum Systems with Conditional Generative Models. arXiv preprint arXiv:2211.16943.

**Questions:**

- See "Weakness" section above.
- In the section concerning the prediction of correlation functions for the anisotropic Heisenberg model, it would be beneficial if the authors could include an analysis of the prediction MSE for correlation functions associated with two quantum sites at **different** distances.

---

> ### Author Response · Authors · 2023-11-16
> **Response to Reviewer TiJ4**
>
> Thanks for your precious time and considerate suggestions to improve our manuscript.
>
> **Q1: While the authors emphasize the unsupervised pre-training as a key novelty of their proposed method, I just feel that the approach they employ in the pre-training phase, involving the regeneration of measurement results, follows a similar logic to that found in [1]. In [1], the model generates measurement results from incomplete measurement results. Although the proposed model here incorporates additional input information, such as physical conditions, I still believe the authors should cite this previous work.**
>
>
> **A1:** Thank you for the insightful comments. We have cited the suggested paper [1].
>
> In addition, we would like to highlight some differences in the pre-training methodology between our work and that presented in paper [1].
>
> In [1], the pre-training process involves gathering diverse measurement strings ($\mathbf{m}$) and the corresponding reliable frequencies/distributions ($p_{true}$) obtained through post-processing. The loss function is primarily designed to reconstruct the distribution $p_{true}$ for new measurement strings.
>
> In contrast, our approach does not rely on such assumptions. The loss function we designed directly corresponds to the reconstruction of the joint probability distribution of the wave function (Eq.3). Therefore, we only require the measurement string (along with some other easily obtainable physical condition features). This design facilitates real-time adjustments to the pre-trained model without the need for calculating $p_{true}$, contributing to a more responsive adaptation process.
>
>
>
>
>
>
>
> **Q2: Furthermore, the fine-tuning phase of the model shares similarities to the framework described in [2]. Despite the results demonstrating the superiority of the proposed method in Table 1, it would be valuable for the authors to engage in a more in-depth discussion regarding the distinctions between their model and this prior work. Given that Q-TAPE without pretraining also outperforms NN-Classical Shadow, it would be appreciated if the authors could provide further analysis or intuitive explanations for the results to highlight the novelty of the proposed model.**
>
>
> **A2:** Thanks for your valuable question. We would like to clarify the distinctions between our approach and the method outlined in reference [2], and we have supplemented this clarification in the new version of the paper.
>
> While both Q-TAPE and NN-Classical Shadow employ a similar transformer structure in their pipelines, the purpose of the transformer differs.
>
> In Q-TAPE, it functions as an encoder, whereas in NN-Classical Shadow, it resembles more of a decoder. The fact that Q-TAPE, even without pre-training, can to some extent surpass NN-Classical Shadow can be subjectively attributed to its **end-to-end** model nature. The parameters in Q-TAPE are specifically optimized for corresponding objectives (such as quantum phase of matters and correlation function), which often leads to superior performance.
>
> On the other hand, NN-Classical Shadow is essentially a **two-step** model, lacking a conventional fine-tuning phase. The transformer in NN-Classical Shadow primarily serves as a generative model used to reconstruct the quantum state and generate measurements records conditioned on specific physical variables. The transformer is followed by the use of classical shadow to predict specific quantum properties. Thus, it is highly possible that the parameters in the model may not be optimal for the specific task.
>
>
>
> **Q3: In the section concerning the prediction of correlation functions for the anisotropic Heisenberg model, it would be beneficial if the authors could include an analysis of the prediction MSE for correlation functions associated with two quantum sites at different distances.**
>
>
> **A3:** Thank you for your question. For the task of predicting the correlation function, our model outputs a symmetric matrix ($H$) of size $L × L$ for a given training sample. Here, $H_{ij}$ represents the predicted correlation function between the ith and jth qubits. Assuming the true correlation function is denoted as $H'$, the MSE is $1/L^{2} \sum_{ij}(H_{ij}-H^{'}_{ij})^2$. And the loss is the root squared value averaged over the batch samples. This incorporates the correlation functions associated with two quantum sites at different distances.
>
> The explanation above has been metioned in Section 4.2, and we will provide a more explicit explanation in the new revised version.

---

> > ### Comment · Reviewer_TiJ4 · 2023-11-17
> >
> > I thank the authors for clarifying my questions; it is very helpful. I've updated the rating.

---

> > > ### Author Response · Authors · 2023-11-17
> > > **Response to Reviewer TiJ4**
> > >
> > > Gratitude for the upgraded score! Your insights greatly contributed to the paper's improvement. Thank you!

---

### Official Review · Reviewer_2sNf · 2023-10-30

**Soundness:** 3 good
**Presentation:** 3 good
**Contribution:** 3 good
**Rating:** 8
**Confidence:** 4

**Summary:**

There is a lot of hope for using machine learning to advance quantum physics. Many supervised machine learning algorithms have been studied and proposed for important foundational problems in quantum physics, such as classifying quantum phases of matter and predicting properties of quantum systems. Inspired by the power of unsupervised models (such as LLMs and pre-trained models in CV), the authors present a pre-trained model Q-TAPE for quantum physics. The authors showed that the pre-trained model enables strong performance with limited training data, which can be very useful as large-scale experimental data are hard to obtain.

**Strengths:**

Machine learning for quantum many-body physics is an important research topic that has been subject to extensive studies in recent years. As of now, a good pre-trained model for quantum physics problems is unavailable. This work could initiate a fruitful line of research to build a powerful pre-trained model for quantum many-body physics.

The extensive experiments conducted in this work showcase the potential for using a good pre-trained model to significantly reduce the amount of experimental data needed. For example, in the largest system size, L = 31, and smallest experimental data of Fig. 3, the performance improved by >15% using the pre-trained model.

**Weaknesses:**

The authors only considered two downstream tasks (classifying phases and predicting properties).
More tasks would make the work even stronger.

The work is purely empirical (which I find to be a minor issue for this direction of research).

**Questions:**

I can understand how language has many universal features, which makes pre-training very useful. However, I don't yet see what kind of features in quantum physics problems are "universal." Could the authors provide some visualizations of the pre-trained model to showcase what kind of universal features are learned by Q-TAPE? Could the authors also provide more discussions (and possibly an example) to illustrate why pre-training is useful in quantum physics? I believe the authors' argument that pre-training can be very useful in quantum physics, but I am not entirely sure how.

As noted in the Limitations paragraph, the work would be stronger if the authors could provide experiments for a few more downstream tasks. I think predicting entanglement entropy should not be too hard, and I would like to see if pre-training helps there.

It would also be interesting to see for what downstream tasks the pre-trained model Q-TAPE is not particularly helpful now. From the perspective of building along this line of thinking, I think illustrating the shortcomings of the current model is actually very useful.

There are some typos in the Limitations paragraph in Section 5: "... for expereiments.Though".

---

> ### Author Response · Authors · 2023-11-16
> **Response to Reviewer 2sNf (Part 1)**
>
> Thanks for your time and valuable comments. Below we respond to your concerns and hope we can clarify some questions.
>
> **Q1: I can understand how language has many universal features, which makes pre-training very useful. However, I don't yet see what kind of features in quantum physics problems are "universal." Could the authors provide some visualizations of the pre-trained model to showcase what kind of universal features are learned by Q-TAPE? Could the authors also provide more discussions (and possibly an example) to illustrate why pre-training is useful in quantum physics? I believe the authors' argument that pre-training can be very useful in quantum physics, but I am not entirely sure how.**
>
>
> **A1:** Thank you for the reviewer's questions. We hope the following description provides some intuitive insights. In quantum mechanics, the wave function serves as a fundamental mathematical symbol **universally** used to characterize quantum physical systems. Mathematically, it can be understood as a **joint probability distribution** (Eq.1) constituted by all possible states of quantum bits.
>
> During the pre-training phase, **our optimization objective is to reconstruct the joint probability distribution based on the post distribution of the random variables (i.e., Measurement Records)**. This approximation aims to make the model's output consistent with such a joint probability distribution, by employing the **average negative log-likelihood loss** (Eq.3). Additionally, we consider the need for using the same model to approximate the joint distribution of a class of quantum systems. Therefore, we impose conditions on the joint distribution (i.e., physical variable $\mathbf{c}$), ultimately forming the final optimization objective (Eq.2 and Eq.3).
>
> In a word, the joint probability distribution, as a universal symbol characterizing quantum physical systems, allows Q-TAPE to approximate joint probability distributions corresponding to different quantum systems in the final  softmax layer in the pre-training phase. The intermediate layers can then learn hidden patterns of quantum systems, which can be transferred to other downstream tasks.

---

> ### Author Response · Authors · 2023-11-16
> **Response to Reviewer 2sNf (Part 2)**
>
> **Q2: As noted in the Limitations paragraph, the work would be stronger if the authors could provide experiments for a few more downstream tasks. I think predicting entanglement entropy should not be too hard, and I would like to see if pre-training helps there.**
>
>
>
> **A2:** Thank you for the reviewer's suggestion. We have supplemented an additional downstream task: predicting the second-order Rényi entanglement entropy $-\log(\mathrm{tr}(\rho_A^2))$ for the Heisenberg model in the revised version, where $A$ is the left-half subsystem with system size $L/2$ of the $L$-qubit quantum system. The number of training size is set to be $N_t=90$ and the predicted RMSE results are given as folllows.
>
>
> |                         |   $L=8$  |           |          |           |            |  $L=10$  |           |          |           |            |  $L=12$  |           |          |           |            |
> |-------------------------|:--------:|:---------:|:--------:|:---------:|:----------:|:--------:|:---------:|:--------:|:---------:|:----------:|:--------:|:---------:|:--------:|:---------:|:----------:|
> |                         | $K_f=64$ | $K_f=128$ | $K_f=256$ | $K_f=512$ | $K_f=1024$ | $K_f=64$ | $K_f=128$ | $K_f=256$ | $K_f=512$ | $K_f=1024$ | $K_f=64$ | $K_f=128$ | $K_f=256$ | $K_f=512$ | $K_f=1024$ |
> | Classical Shadow        | 1.58282  |  1.56688  | 1.50989  |  1.40270  |  1.22974   | 1.72379  |  1.71451  | 1.73135  |  1.72740  |  1.68556   | 2.89481  |  2.90874  | 2.91391  |  2.90773  |  2.89722   |
> | RBF Kernel              | 0.07322  |  0.07160  | 0.07670  |  0.07692  |  0.07706   | 0.02539  |  0.02257  | 0.02242  |  0.02002  |  0.01983   | 0.08710  |  0.08242  | 0.08104  |  0.07081  |  0.07032   |
> | NTK                     | 0.07117  |  0.06799  | 0.08834  |  0.08708  |  0.08690   | 0.02497  |  0.02221  | 0.02129  |  0.01996  |  0.01947   | 0.08432  |  0.08249  | 0.08071  |  0.07998  |  0.07381   |
> | PixelCNN                | 0.07198  |  0.07091  | 0.06849  |  0.06687  |  0.06784   | 0.01907  |  0.01892  | 0.01948  |  0.01952  |  0.02089   | 0.07406  |  0.07145  | 0.07107  |  0.06895  |  0.06677   |
> | Neural-Classical Shadow | 0.06860  |  0.06415  | 0.06403  |  0.06315  |  0.06221   | 0.01844  |  0.01747  | 0.01664  |  0.01662  |  0.01657   | 0.07261  |  0.06858  | 0.06573  |  0.06156  |  0.05924   |
> | Q-TAPE                  | **0.06302**  |  **0.06141**  | **0.06104**  |  **0.05998**  |  0.06072   | **0.01698**  |  **0.01623**  | **0.01534**  |  **0.01517**  |  **0.01520**   | **0.05861**  |  **0.05812**  | **0.05648**  |  **0.05623**  |  **0.05597**   |
> | Q-TAPE w/o Pretrain     | 0.06649  |  0.06295  | 0.06228  |  0.06071  |  **0.06034**   | 0.01711  |  0.01662  | 0.01696  |  0.01655  |  0.01532   | 0.06624  |  0.06542  | 0.06381  |  0.06042  |  0.05931   |
>
> It can be observed that pre-training remains effective for predicting the entanglement entropy of the anisotropic Heisenberg model. We hope that these additional experimental results will provide the reviewers with further insights.
>
> **Q3: It would also be interesting to see for what downstream tasks the pre-trained model Q-TAPE is not particularly helpful now. From the perspective of building along this line of thinking, I think illustrating the shortcomings of the current model is actually very useful.**
>
>
> **A3:** Thank you for the insightful suggestions. In the Limitation section, we acknowledged some current limitations of our model. Due to space constraints, these discussions were not exhaustive. Here, we attempt to provide further elaboration.
>
> We acknowledge that the current model may not effectively characterize quantum systems controlled by the time-dependent Hamiltonian. The proposed approach in this paper is currently limited to the estimation of quantum properties for Hamiltonians that are time-independent. We believe that exploring how to leverage pre-training methods from traditional deep learning for handling time-series data could be a promising direction for investigating the quantum time evolution of many-body systems.
>
> We appreciate the reviewer's attention to this aspect, and we agree that addressing the challenges associated with time-dependent Hamiltonians in quantum systems is an avenue worth exploring in future research.
>
>
> **Q4: There are some typos in the Limitations paragraph in Section 5: "... for expereiments. Though".**
>
> **A4:** Thank you for the reviewer's guidance. We have incorporated the suggested corrections in the revised version of the paper.

---

> > ### Comment · Reviewer_2sNf · 2023-11-21
> >
> > I really appreciate the effort by the authors to strengthen the paper further.
> > This is a good paper, so I will keep my score of 8.

---

> > > ### Author Response · Authors · 2023-11-22
> > > **Response to Reviewer 2sNf**
> > >
> > > Thanks for your reply! Your feedback significantly enhanced the paper. Hope you enjoy your Thanksgiving!

---

### Official Review · Reviewer_qDBp · 2023-10-30

**Soundness:** 3 good
**Presentation:** 3 good
**Contribution:** 3 good
**Rating:** 8
**Confidence:** 5

**Summary:**

In this work, the authors leverage a pre-trained LLM strategy to construct the Q-TAPE architecture to deal with quantum data, which involves an unsupervised pre-trained model and fine-tuning for specific tasks. Numerous experiments have been conducted to show the promising efficacy of the Q-TAPE.

**Strengths:**

1. The work leverages the pre-training strategy that has attained triumph in large language models.

2. This work aims at building quantum datasets that can be conceptually equivalent to the corpus used to train LLMs.

**Weaknesses:**

1. The idea of generating and collecting quantum data is important, but quantum data is more fitted to quantum machine learning models like quantum neural networks and quantum graph neural networks, instead of the classical ones.

2. Even though the work somehow makes the generated quantum data suitable for the classical LLM architecture, the new hybrid quantum-classical architecture is not interesting. For classical-to-quantum data conversion, a simple quantum tensor encoding, amplitude encoding, or a more advanced method of the quantum kernel can efficiently deal with quantum data generation. If you could collect quantum datasets, the use of quantum machine learning models is more important and that is the most advantage of quantum neural networks.

3, Similar techniques of pre-training strategy have been shown in many quantum machine learning works. For example, Yang's quantum NLP work has demonstrated the effectiveness of pre-training hybrid LLM for dealing with quantum data. Moreover, Qi's recent work has provided a solid theoretical understanding of pre-training LLM for quantum machine learning. Unfortunately, those important and very related works are not cited in this work.

Ref. [1] Yang, C.H.H., Qi, J., Chen, S.Y.C., Tsao, Y. and Chen, P.Y., 2022, May. When Bert meets quantum temporal convolution learning for text classification in heterogeneous computing. In IEEE International Conference on Acoustics, Speech and Signal Processing (pp. 8602-8606). IEEE.

Ref. [2] Jun Qi, Chao-Han Huck Yang, Pin-Yu Chen, Min-Hsiu Hsieh, "Pre-Training Tensor-Train Networks Facilitate Machine Learning with Variational Quantum Circuits," arXiv:2306.03741v1.

4. In the experiments, the authors do not discuss the in-distribution and out-of-distribution quantum data. For the out-of-distribution quantum data, the proposed pre-training LLM architecture cannot ensure a good performance. So, some new quantum-aware optimization algorithms need to be more deeply investigated.

**Questions:**

(1) Why not directly use quantum machine learning like quantum neural networks to deal with the quantum datasets? since the hybrid quantum-classical architecture can attain competitive empirical results.

(2) Is there some out-of-distribution quantum data being existed in the datasets? If so, how to deal with the case.

---

> ### Author Response · Authors · 2023-11-16
> **Response to Reviewer qDBp (Part 1)**
>
> Thank you for the time, thorough comments, and nice suggestions. We are pleased to clarify your questions step-by-step.
>
> **Q1: The idea of generating and collecting quantum data is important, but quantum data is more fitted to quantum machine learning models like quantum neural networks and quantum graph neural networks, instead of the classical ones.**
>
> **A1:** We appreciate the concerns raised by the reviewer, specifically regarding the suitability of quantum approaches to address quantum problems. We have also noted the increasing use of quantum variational circuit algorithms for handling classical and quantum tasks.
>
> However, **we hold the view that scalable fault-tolerant quantum computers, while promising for solving a wide range of quantum problems, are unlikely to be available in recent years [1].**  Moreover, whether the expressive and generalization capabilities of quantum variational circuit algorithms are stronger than the traditional deep learning algorithms, remain an open question.
>
> Classic deep learning methods, having demonstrated successfully in various domains such as text, images, and physics, are now being extended to characterize quantum many-body systems [1,2,3,4]. **In line with this trend, we clarify the basic motivation behind our paper: how can we best leverage our powerful classical computers to enhance our comprehension of complex quantum systems?**
>
> As such, we define the strategy for generating and collecting essential data types characterizing quantum systems on quantum computers or simulators. We also outline a pre-training model Q-TAPE that utilizes this collected data to train a classical machine learning model capable of predicting certain features of unknown or even quantum systems that may pose challenges for current quantum computers. The experimental evidences demonstrate the superiority of Q-TAPE in classifying quantum phase of matters and predicting correlation functions.
>
> We hope that such classical model could inspire more reasearch on investigating the pre-trained mechanism on sovling quantum many body problems. We also hope the proposed model could provide reliable assistance for the verification and certification of quantum devices.
>
>
> [1] Huang H Y, Kueng R, Torlai G, et al. Provably efficient machine learning for quantum many-body problems[J]. Science, 2022, 377(6613): eabk3333.
>
> [2] Carleo G, Troyer M. Solving the quantum many-body problem with artificial neural networks[J]. Science, 2017, 355(6325): 602-606.
>
> [3] Wu D, Wang L, Zhang P. Solving statistical mechanics using variational autoregressive networks[J]. Physical review letters, 2019, 122(8): 080602.
>
> [4] Gebhart V, Santagati R, Gentile A A, et al. Learning quantum systems[J]. Nature Reviews Physics, 2023, 5(3): 141-156.

---

> > ### Comment · Reviewer_qDBp · 2023-11-16
> > **Good responses to my concerns**
> >
> > I thank the authors' feedback on my concerns about the work, and their answers best handle all of the raised comments. Then, I would like to change my evaluation scores for this paper.

---

> > > ### Author Response · Authors · 2023-11-17
> > > **Response to Reviewer qDBp**
> > >
> > > Thanks for your quick reply! Your feedback was invaluable for us!

---

> ### Author Response · Authors · 2023-11-16
> **Response to Reviewer qDBp (Part 2)**
>
> **Q2: Even though the work somehow makes the generated quantum data suitable for the classical LLM architecture, the new hybrid quantum-classical architecture is not interesting. For classical-to-quantum data conversion, a simple quantum tensor encoding, amplitude encoding, or a more advanced method of the quantum kernel can efficiently deal with quantum data generation. If you could collect quantum datasets, the use of quantum machine learning models is more important and that is the most advantage of quantum neural networks.**
>
> **A2:** Thank you for the reviewer's response. We apologize for any lack of clarity. We will try our best to provide a detailed explanation below (and also in our paper).
>
> Our model, Q-TAPE, is an **entirely classical machine learing model** and is designed to address quantum many-body problems (including classifying quantum phase of matters and predicting the correlation matrix of quantum systems).
>
> In our paper, **we define the "quantum data" required for training the Q-TAPE to handle quantum many-body related problems**. This data is obtained through **multiple measurements on identical copies of a "quantum" system**. The collected data is "classical", capable of being stored on disk and in memory for training and testing Q-TAPE. **This process involves only quantum-to-classical transformations, such as measurements through POVM, and does not involve classical-to-quantum data conversion.**
>
> As mentioned in response to Question 1, quantum models (including quantum neural networks and quantum kernel methods mentioned by the reviewer) inherently possess advantages in handling quantum problems and quantum data. At least, there is no need for us to convert quantum data into classical data, which often incurs significant computational and experimental costs.
>
> **However, considering the current limitations of quantum computers—small-scale (a few hundred qubits), high noise, and short coherence times—it is often impractical to use such quantum computers to solve real-world problems. Even when these quantum algorithms run in classical simulation environments, the scale of simulation is generally limited to toy models.** Therefore, employing purely classical machine learning methods to address quantum problems remains a viable consideration. We hope that our proposed Q-TAPE provides a new perspective on pre-training methods for solving quantum many-body problems.
>
> **Q3: Similar techniques of pre-training strategy have been shown in many quantum machine learning works. For example, Yang's quantum NLP work has demonstrated the effectiveness of pre-training hybrid LLM for dealing with quantum data. Moreover, Qi's recent work has provided a solid theoretical understanding of pre-training LLM for quantum machine learning. Unfortunately, those important and very related works are not cited in this work.**
>
> **A3:** Thank you for your consideration. We carefully reviewed these valuable works mentioned by the reviewer (Yang et al. and Qi et al.) and we also have appropriately cited them in our revised paper.
>
> Although both our paper and the papers mentioned by the reviewer include the term "quantum", it is crucial to emphasize that, to a large extent, our motivations and methodology are not directly relevant to the papers mentioned by the reviewer. The reasons include:
>
> 1. Firstly, we note that both two papers focus on using hybrid models to address classical problems, with Yang et al. concentrating on text classification and Qi et al. on image classification. **In contrast, our model is entirely classical and developed to tackle quantum problems.**
> 2. Secondly, in our paper, the term "**quantum data**" refers to **measurement data obtained from quantum systems through random Pauli measurements**, preserving the characteristics of the quantum system. It is essential to note that in the mentioned papers, "quantum data" specifically refers to **quantum embedding**, which involves encoding classical data through quantum circuits, essentially creating a **quantum representation of classical data**.
> 3. While Yang et al. and our model both incorporate components of large language models (LLMs), Yang et al.'s motivation lies in **utilizing a hybrid model to address classical problems, specifically text classification. In their approach, a quantum neural network serves as an additional layer to enhance the expressive power of the traditional BERT model.** On the other hand, **our proposed Q-TAPE is designed to characterize and predict features of quantum systems without the involvement of any quantum models.** Q-TAPE achieves this by unsupervised pre-training, learning hidden patterns of quantum systems, and utilizing these patterns during fine-tuning to achieve better results with fewer measurements.

---

> ### Author Response · Authors · 2023-11-16
> **Response to Reviewer qDBp (Part 3)**
>
> **Q4: In the experiments, the authors do not discuss the in-distribution and out-of-distribution quantum data. For the out-of-distribution quantum data, the proposed pre-training LLM architecture cannot ensure a good performance. So, some new quantum-aware optimization algorithms need to be more deeply investigated.**
>
> **A4:** Thank you for the reviewer’s comments. We apologize for not being specific and clear about the disrtibution of quantum data we use in the paper. In our context, the quantum data used for pre-training and fine-tuning **comes from the same distribution**. We consider the work of handling out-of-distribution data as future work. Even though, we have conducted some preliminary experiments related to OOD data, hoping to better give the reviewer more insight of our model.
>
> From our understanding of the review, the OOD dataset mentioned by the reviewer refers to the dataset used for fine-tuning and the dataset used for pre-training from different distributions.
>
> Here, we consider **two different configurations** to make the fine-tuning dataset out-of-distribution from the pre-training one: the first is to re-generate the fine-tuning data by **modifying the physical variables** and the second is to fine tune the model based on the parameters **transferred from the model pretrained on fewer qubits**. In the following, we consider the Rydberg model. The training size is set to be $N_t=100$, and the number of measurements $K_f=1024$. The details are as follows:
>
> First, we take the evaluation that fine-tuning the model on 31-qubit system by using he parameters pre-trained on 19 and 25-qubit system. Note that the number of qubits is also a physical variable and we want to see if model parameters trained on small-scale systems could transfer and help model characterize larger-scale systems. The results are listed below:
>
> |  method                             | accuracy |
> | --------------------------------------- | -------- |
> | Q-TAPE (pre-trained on 19-qubit system) | 95.74    |
> | Q-TAPE (pre-trained on 25-qubit system) | 96.13    |
> | Q-TAPE (pre-trained on 31-qubit system) | 96.67    |
> | Q-TAPE w/o pre-train                    | 94.32    |
>
>
> From the above table, it is evident that pre-trained parameters transferred from small-scale systems is also useful for large-scale systems.
>
> Second, we modify the detuning of a laser from [-10,15] (which is exactly used in the paper) to $[-20,-10]\cup [15,25]$ to generate OOD fine-tuning dataset, on Rydberg atom model with 19 qubits. The classification accracy are listed below:
>
> | method             | no OOD | OOD   |
> | -------------------- | ------ | ----- |
> | Q-TAPE               | 95.95  | 84.82 |
> | Q-TAPE w/o pre-train | 93.35  | 94.23 |
>
> where the *no OOD* denotes that the fine-tuning dataset are generated from detuing [-10,15] as the same as the pre-training.  The pre-trained one fails to perform better than the Q-TAPE w/o pre-train. The main reason is that the modified detuning values habe driven the quantum evolution into a very different dynamics and the pre-trained model learns less knowledge about it.
>
> The question of whether pre-trained Q-TAPE remains beneficial for OOD quantum datasets in other settings remains open, and will be further explored in our future work.
>
> We hope these replies help. Please let us know if you have any further questions for the OOD problems.
>
> **Q5: Why not directly use quantum machine learning like quantum neural networks to deal with the quantum datasets? since the hybrid quantum-classical architecture can attain competitive empirical results.**
>
> **A5:** The reason why we did not directly use quantum machine learning like quantum neural network has been explained in Q1 and Q2. We are more than willing to address any further inquiries or comments to ensure clarity and understanding.
>
> Additionally, the reviewer mentioned that hybrid methods can attain competitive empirical results. However, we **did not** find relevant empirical results in the cited references (Yang et al. and Qi et al.) given by the reviewer, and these references differ significantly from the specific task we are addressing. If the reviewer could provide additional literature for comparison, we would be pleased to assess and discuss the results further.
>
>
> **Q6: Is there some out-of-distribution quantum data being existed in the datasets? If so, how to deal with the case.**
>
> **A6:** This has been discussed in Q4. Please refer to our answer in Q4.

---

### Official Review · Reviewer_1o9o · 2023-10-31

**Soundness:** 3 good
**Presentation:** 3 good
**Contribution:** 3 good
**Rating:** 8
**Confidence:** 5

**Summary:**

The submission introduces Q-TAPE, a versatile pre-trained model for quantum systems, aiming to enhance property estimation. Drawing inspiration from Large Language Models' success in other domains, Q-TAPE offers several key advantages: it leverages a rich set of quantum data for comprehensive training, adopts an unsupervised and task-agnostic approach, excelling in classifying quantum phases and predicting correlation functions, especially with limited data, and seamlessly transfers knowledge from pre-training to specific properties estimation tasks. Extensive experiments validate Q-TAPE's efficacy across various quantum tasks. Furthermore, the authors commit to making the source code openly available, fostering further research and applications in quantum property estimation.

**Strengths:**

Q-TAPE presents a novel and promising approach to understanding quantum systems. Diverging from previous methods that primarily rely on supervised learning, Q-TAPE advocates an unsupervised pretraining combined with downstream task finetuning, a methodology akin to the training of large language models. This paradigm shift offers a more experimentally-friendly approach, emphasizing adaptability and robustness. The impressive results achieved, especially with quantum systems involving up to 31 qubits, underscore the significant potential of Q-TAPE, solidifying its strengths in advancing quantum property estimation.

**Weaknesses:**

Several minor concerns about the submission are listed below.

Firstly, it is essential to empirically investigate the scaling behavior of the required number of measurements for each training example concerning the qubit count. A polynomial scaling that ensures satisfactory performance is desirable, as an exponentially scaling behavior may impede the practical utility of Q-TAPE for larger quantum systems.

Secondly, there is a need for a discussion on the storage method employed for the training dataset and the corresponding memory cost. For larger qubit counts, as the number of measurements increases, the memory cost may become prohibitively expensive even though only the measurement results and the measured operators are stored. It would be beneficial to know if the authors have developed advanced methods to address this challenge, as the bottleneck of employing deep learning for quantum problems could shift to data management.

Lastly, some missing references that are pertinent to the content of the submission should be discussed, ensuring a comprehensive and well-referenced presentation of the work. Concrete examples include Refs [1] and [2], which separately address the ability of deep neural networks to simultaneously accomplish multiple tasks and the fundamental role of datasets in quantum system learning.

[1] Wu, Ya-Dong, et al. "Learning and Discovering Quantum Properties with Multi-Task Neural Networks." arXiv preprint arXiv:2310.11807 (2023).
[2] Du, Yuxuan, et al. "ShadowNet for Data-Centric Quantum System Learning." arXiv preprint arXiv:2308.11290 (2023).

**Questions:**

The questions are listed in Weaknesses.

---

> ### Author Response · Authors · 2023-11-16
> **Response to Reviewer 1o9o (Part 1)**
>
> Thanks for your constructive suggestions. Your endorsement of our method and experiments gives us significant encouragement. Here are our clarifications.
>
> **Q1: Firstly, it is essential to empirically investigate the scaling behavior of the required number of measurements for each training example concerning the qubit count. A polynomial scaling that ensures satisfactory performance is desirable, as an exponentially scaling behavior may impede the practical utility of Q-TAPE for larger quantum systems.**
>
> **A1:** Thank you for your valuable feedback. We acknowledge the significance of investigating the scaling issues related to the Q-TAPE. While we did mention some aspects of the model's scaling in the main text, we recognize that it may not have been explicitly emphasized. In this response, we aim to provide a detailed explanation below and we have added them in our new revised version.
>
> It is crucial to emphasize that **the exponential demand for measurements constitutes a distinctive gap between quantum and classical methods for completing quantum tasks, such as classifying phases and estimating non-linear properties [1]**.
>
> Even though, Q-TAPE focuses more on practical applicability in quantum properties estimation tasks, assuming a sufficient number of measurement outcomes are given. Correspondingly, it is empirically evident in Fig. 3 that achieving linear growth in classification accuracy requires an exponential increase in the number of measurements per training example.
>
> Notably, **the advantage lies in the potential value of the pre-training of Q-TAPE, particular in scenarios where only a limited number of measurements are available.**

---

> ### Author Response · Authors · 2023-11-16
> **Response to Reviewer 1o9o (Part 2)**
>
> Additionally, beyond the scaling related to number of measurements, we dive into further research on the scaling relationship between accuracy and **the size of the training set** (i.e., the number of sampled physical conditions which determine the dynamics of the quantum system). We constrain the number of measurement per example to 256 (since we find that a large value makes the accuracy reach saturation) and the results on the 31-qubit system are listed in the table below. The results are averaged over 3 runs with different random seeds.
>
>
> |                     | $N_t=20$ | $N_t=40$ | $N_t=60$ | $N_t=80$ |
> | ------------------- | -------- | -------- | -------- | -------- |
> | Q-TAPE              | 82.05    | 87.24    | 89.16    | 90.63    |
> | Q-TAPE w/o pretrain | 79.17    | 81.78    | 85.96    | 88.47    |
>
>
> As evident from the table, accuracy approximately exhibits **linear** growth w.r.t. training size. This finding consistents with theoretical results presented in [2,3], which demonstrate that there exists a **polynomial scaling** relationship between model performance and the size of training dataset.
>
> We appreciate your insightful comments, and we hope this clarification could address your concerns.
>
> [1] Huang H Y, Broughton M, Cotler J, et al. Quantum advantage in learning from experiments [J]. Science, 2022, 376(6598): 1182-1186.
>
> [2] Huang H Y, Kueng R, Torlai G, et al. Provably efficient machine learning for quantum many-body problems [J]. Science, 2022, 377(6613): eabk3333.
>
> [3] Lewis L, Huang H Y, Tran V T, et al. Improved machine learning algorithm for predicting ground state properties [J]. arXiv preprint arXiv:2301.13169, 2023.

---

> ### Author Response · Authors · 2023-11-16
> **Response to Reviewer 1o9o (Part 3)**
>
> **Q2: Secondly, there is a need for a discussion on the storage method employed for the training dataset and the corresponding memory cost. For larger qubit counts, as the number of measurements increases, the memory cost may become prohibitively expensive even though only the measurement results and the measured operators are stored. It would be beneficial to know if the authors have developed advanced methods to address this challenge, as the bottleneck of employing deep learning for quantum problems could shift to data management.**
>
> **A2:** Thank you for your feedback. To be honest, in the experiments conducted for this paper, we did not consider advanced storage methods for efficiently managing resources while collecting measurement outcomes of quantum systems. Instead, we packaged them in tensor form (e.g., measurement records represented as a K×L matrix) and saved them as .npy files. The .npy files for each training example that were, at most, a few tens of MB, which we think is acceptable.
>
> Nevertheless, recognizing that Q-TAPE is designed to handle larger quantum datasets generated from large-scale quantum systems in the future, here we provide two forward-looking suggestions that could be effective for storing these measurement results. We are open to further discussion with the reviewers to explore even more optimal solutions for the storage of quantum measurement results:
>
> 1. Utilizing a Key-Value Pair Approach:
> Employing a key-value pair format such as {measurement string: frequency} for storage could be beneficial. This approach avoids redundant storage of identical measurement strings, optimizing the storage space.
>
> 2. Implementing Huffman Trees (or other tree-based data structures):
> Using Huffman trees or similar tree-based data structures to store measurement strings could be considered. Placing the most frequently occurring measurement strings on the shortest paths of the Huffman tree ensures efficiency in both storage and retrieval when training the model.
>
>
> **Q3: Lastly, some missing references that are pertinent to the content of the submission should be discussed, ensuring a comprehensive and well-referenced presentation of the work. Concrete examples include Refs [1] and [2], which separately address the ability of deep neural networks to simultaneously accomplish multiple tasks and the fundamental role of datasets in quantum system learning.**
>
> **A3:** Thank you for your suggestions. We have incorporated the missing references into the paper.

---

> ### Comment · Reviewer_1o9o · 2023-11-22
>
> I appreciate the authors' detailed response. They have adequately addressed most of my concerns. Consequently, I have decided to raise my score to 8.

---

> > ### Author Response · Authors · 2023-11-22
> > **Response to Reviewer 1o9o**
> >
> > Many thanks for the score improvement! Your feedback greatly enriched the paper. Hope you enjoy your Thanksgiving!

---

### Meta-Review · Area_Chair_sNfW · 2023-12-04

**Metareview:**

This paper studies the problem of quantum properties estimation via machine learning approach, in particular inspired by LLMs and pre-trained models. The proposed model, Q-TAPE, can learn information from diverse quantum systems with different physical conditions in a fully unsupervised fashion, and can deliver high performance with limited training data, mitigating the cost for quantum data collection and reducing the time for convergence for different supervised tasks. Numerical experiments demonstrate the promising efficacy of Q-TAPE in various tasks including classifying quantum phases of matter on Rydberg atom model and predicting two-body correlation function on anisotropic Heisenberg model.

There are notable strengths of this paper:
- The capability and performance is impressive with quantum systems involving up to 31 qubits, and the performance improved by >15% using the pre-trained model.
- The technique of fine tuning in LLM is mainstream with wide impact at the moment, and it's nice to see that it can be extended to solving physics problems.
- Moreover, the problems studied by the paper, including quantum phases of matter on Rydberg atom model and predicting two-body correlation function on anisotropic Heisenberg model, are important in many-body physics.

There are also weaknesses of the paper, mainly relatively narrow experiments and lack of discussing about relevant literatures in the initial version. In the rebuttal, the authors had made significant efforts in addressing these issues, and the reviewers found the changes satisfactory with scores increased.

As the rebuttal had very detailed discussions with many new experiment results, in the final version the authors should make significant changes by merging the rebuttals into the paper.

**Justification For Why Not Higher Score:**

I'm not very sure if the topic of solving quantum physics problems by fine tuning LLMs has the most general audiences of ICLR 2024 and qualifies for the very competitive oral slots. The applications are more on the quantum side and the technique in terms of machine learning are not super novel.

**Justification For Why Not Lower Score:**

In general, AI for Science is a main trend for machine learning research, and it's nice to see that this submission can make notable contributions - it can solve quantum physics problems with 31 qubits (i.e. system of dimension 2^31), with improved performance by the pre-trained model. This result should be a very good example among AI for Science, and I believe that it should not only be a poster but a spotlight. The average score of 8 from the reviewers also agrees with this.

---

### Decision · Program_Chairs · 2024-01-16

Accept (spotlight)